# Specialization of Business Process Model and Notation Applications in Medicine—A Review

**Hana Tomaskova ***[ID] **and Martin Kopecky**

Faculty of Informatics and Management, University of Hradec Kralove, Rokitanskeho 62,
50003 Hradec Kralove, Czech Republic; martin.kopecky@uhk.cz

***** Correspondence: hana.tomaskova@uhk.cz

**Abstract:** Process analysis and process modeling are a current topic that extends to many areas. This trend of using optimization and modeling techniques in various specific areas has led to the question of how widespread these approaches are overall in medical specializations. We compiled a list of 272 medical disciplines that we used as a search string with the Business Process Model and Notation (BPMN) for a Web of Science database search. Thus, we found a total of 485 documents that we subjected to the exclusion criteria. We analyzed the remaining 108 articles using bibliometric and content analyses to find answers to three research questions. This systematic review was carried out using the procedure proposed by Kitchenham and following the Preferred Items of the Systematic Review and Meta-Analysis Report (PRISMA). Due to the broad scope of the medical field, it was no surprise that for almost 85% of the sought-after medical specializations, we could not identify any publications in the given database when applying the BPMN. We analyzed the impact of upgrades to the BPMN on publishing. The keyword analysis showed a diametrical difference between the authors' keywords and the so-called "Keywords Plus", and we categorized the publications according to the purpose of applying the BPMN. However, the growing interest in combining BPMN with other approaches brings new challenges in practice.

**Keywords:** BPMN; systematic review; medical; medical specialties; Business Process Model and Notation; Business Process Management

---

## 1. Introduction

At present, the effort and need to integrate information technology (IT) prevails. This is thus becoming the standard, and not only for multi-institutional organizations. The main purpose of automation and system integration is to reduce costs and improve service quality, which is essential in many areas. Increasing interoperability, higher accuracy of medical data, and compliance with new regulations only emphasize the importance of approaches using the Business Process Model and Notation (BPMN). Organizational processes and decision support can be captured in many ways, of course. We can mention dynamic simulation [1–6], strategic management [7–13], economic analyses, or information technology [14–17]. Some authors try to provide a solution for the analysis of process models. For example, References [18,19] discussed the strengths and limitations of the various modeling approaches used in business process transformation. The paper [20] compares three process modeling processes used in case studies. The article [21] analyzes process models using graph reduction techniques. Other authors, such as [22,23], use specific tools, frameworks, and methods for process analysis and modeling [24].

In healthcare, process management generally reduces the time for change, ensures the visibility of the entire management and decision-making lifecycle, and enables an effective response to nonstandard situations [25]. Process management also upgrades or redesigns processes to support new clinical

practices, regulatory standards, cost-recovery methods, and the like. Treatment processes are very complex, and their graphical visualization facilitates their management and improvement. For this reason, this work aims to present successful applications of the BPMN in the management of medical processes.

This text includes the following: After a short introduction, which includes related studies and necessary information about the BPMN, a section on research methodology follows. The third part presents the results of the review. First, we offer a summary of the primary data about the research sample. Subsequently, we perform analyses according to the focus on article metrics, sources, author analysis, countries and affiliation, keywords, and content analysis. The article concludes with the limitations of the study and the Conclusions section, where we present a summary of the most important limitations and results.

### 1.1. Related Works

Business process modeling is an essential task in business process management. In this paper [26], the authors conducted a systematic review of the literature, where they identified that few authors have explored elements of user interaction in their works. The purpose of the paper [27] was to study the feasibility of combining the business processes (BP) with agent-based models to improve performance, manage resources, and ensure coordination between them. The authors present multi-agent solutions representing social networks in the healthcare domain associated with a Business Process Management (BPM) of patient pathways.

As discussed below, one of the four exclusion criteria for the study was the designation of the article as a review. We excluded a total of two articles from the study because they were review papers. The first is the paper [28], where the authors aimed to assess the results of the application of Business Process Management as a new strategy for process management to optimize clinical processes. The second document [29], among its results, suggests that the Business Process Model and Notation (BPMN) and Service Component Architecture (SCA) were not generally accepted to affect the performance of IT healthcare systems for better care solutions.

### 1.2. Business Process Model and Notation

In general, we can consider the BPMN as a language for creating business process models or as a standard for modeling business processes. The Business Process Management Initiative (BPMI) made this notation entirely open. The BPMN may seem like flowcharts or Petri nets, but instead, it provides much more sophisticated tools for describing and simulating the behavior as well as greater user friendliness [30,31].

The Business Process Management Initiative (BPMI) created the first version, BPMN 1.0, in 2004. In 2005, the BPMI merged with the Object Management Group (OMG). The following year, the OMG issued the BPMN specification document. The OMG developed BPMN 2.0 in 2010, and the current version, BPMN 2.0.2, was released in December 2013. The history of the BPMN and notation development is a frequent topic of BPMN publications; we can mention [31–36].

The wide-ranging use in process analysis is the primary purpose of this notation. It is intelligible to nonspecialists and, at the same time, the notation allows sharing and conversation between different participants [37].

## 2. Research Methodology

A systematic literature review methodology for multidisciplinary or IT areas is not easy to find. Therefore, we used the article [38], which states that a systematic review of the literature for IT should include three basic things. The first is to determine the research question or the research goal of the whole study. That is followed by an equally important organization of impartial and extensive analyses of related publications, and, thirdly, the establishment of explicit inclusion and exclusion criteria.

After many revisions of partial reviews and after the reviewers' recommendations, we determined the following three main research questions for the current systematic literature review:

1. Research Question (RQ1): Have the different versions of the BPMN brought about a change in its use in the medical domain?
2. Research Question (RQ2): In which medical specialization is the BMPN used?
3. Research Question (RQ3): What type of use is made of the BMPN in these specialities?

The analysis process and criteria are given in the following relevant subsections.

### 2.1. Eligibility Criteria

The primary sample of the study includes the publications listed in the database Web of Science (WOS) of Clarivate Analytics, which were published between 1 January 2004 and 10 June 2020 and that contained the search strings. The year 2004 was selected as the starting time point, as it is the year in which the BPMN was created by the Business Process Management Initiative (BPMI).

As conditions excluding publications from the final review, we established the following exclusion criteria:

1. Exclusion Criterion (EC1): The language of the publication is not English.
2. Exclusion Criterion (EC2): The paper is a review.
3. Exclusion Criterion (EC3): The application of the BPMN is not in the medical field.
4. Exclusion Criterion (EC4): The abbreviation "BPMN" does not mean "Business Process Model and Notation" or "Business Process Modeling Notation".

### 2.2. Information Sources and Search

We chose the Clarivate Analytics Web of Science (WOS) database as the primary data source for the study. This database contains publications that have undergone a review process, which is considered as fundamental in scientific circles, and its outputs provide complete information suitable for analysis. We performed an advanced search for the search queries below. We performed a search for the Topics section (TS) as the broadest content section. In the WOS database, we chose the Core sub-database with the indexes listed in Table 1. We limited the study to "All document types", "All languages", and the years 2004–2020.

**Table 1.** Web of Science Core collection indexes.

| Indexes | Abbreviation |
|---|---|
| Science Citation Index Expanded | (SCI-EXPANDED) |
| Social Sciences Citation Index | (SSCI) |
| Arts and Humanities Citation Index | (A&HCI) |
| Conference Proceedings Citation Index-Science | (CPCI-S) |
| Conference Proceedings Citation Index-Social Science & Humanities | (CPCI-SSH) |
| Book Citation Index-Science | (BKCI-S) |
| Book Citation Index-Social Sciences & Humanities | (BKCI-SSH) |
| Emerging Sources Citation Index | (ESCI) |
| Current Chemical Reactions | (CCR-EXPANDED) |
| Index Chemicus | (IC) |

### 2.3. Study Selection

We divided the screening of publications into several phases. In the first phase, we assessed the titles and abstracts of the documents according to the exclusion criteria mentioned above. We considered the rest of the publications during full-text reading and included two independent evaluators who verified our results. We did not exclude records based on methodological quality.

We further examined the studies that went through the introductory network from many points of view and coded according to various criteria. This review aims to present the current progress in linking medical specialties with the BPMN's modeling notation. The results of this study could encourage researchers to use the BPMN in more medical specializations.

A limitation of this review is that it limits the analysis to English-language publications published from the 1 January 2004 to 10 June 2020 that were found through precise queries. This limitation may have omitted some relevant studies in other languages, those that were published after 10 June 2020, or those with which there was an inability to link the search query with the paper.

*2.4. Data Collection Process*

The medical domain is a vast area, and that is why we have chosen two primary official international documents. These were:

1.　"Directive 2005/36/EC of the European Parliament and of the Council of 7 September 2005, on the recognition of professional qualifications (Text with EEA relevance)" [39] and
2.　"List of specialties, fields of speciality practice, and related specialist titles by the Medical Board of Australia" [40].

With these, we established a list of fundamental medical specializations. This list is set out in Appendix A in Table A1.

For data collection, we used an advanced search in the WOS database. We compiled the appropriate number of search queries for each medical specialization. The raw format is: TS = (BPMN AND "Medical specialization from Table A1"). We further examined the documents to see that they did not meet exclusion criteria EC1–EC4, following the procedure proposed by Kitchenham [38] and according to the Preferred Reporting Items for Systematic Controls and Meta-Analyses (PRISMA) [41]. At least one disqualification response was required to exclude a study record. In case of doubt or inconsistency, the final decision was made by the principal author of this publication. Detailed phases, including quantifiers, were presented in the thematically adapted PRISMA flowchart in Figure 1.

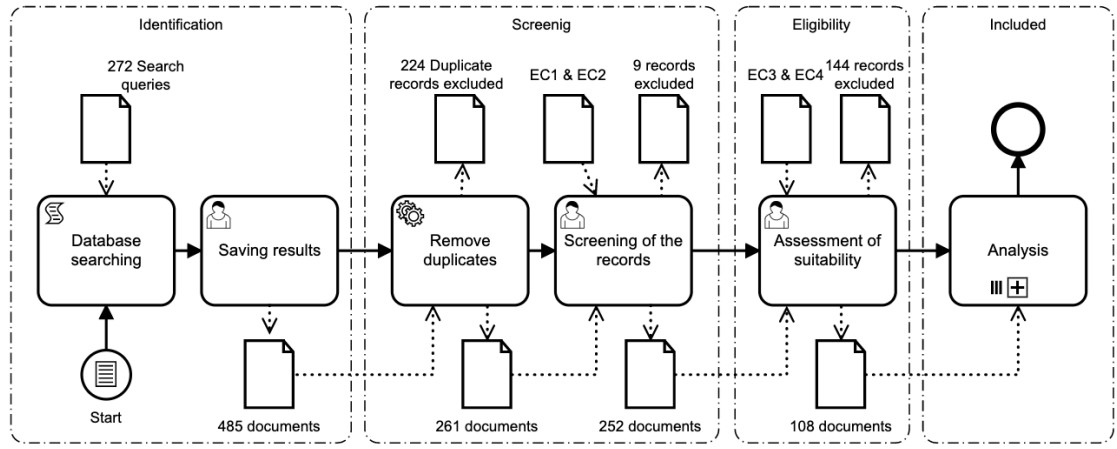

**Figure 1.** Business Process Model and Notation (BPMN) Preferred Reporting Items for Systematic Controls and Meta-Analyses (PRISMA) flowchart; authors' own processing.

The first step involved removing duplicate items. The second step was to use the exclusion criteria (EC1 and EC2). These studies (EC2) are listed in the Related Work section. The evaluation of the studies followed this according to their content and the involvement of EC3 and EC4. Based on the exclusion criteria, the results did not contain the following numbers of publications: EC1 = 7; EC2 = 2; EC3 = 136; EC4 = 8.

Table 2 contains the search strings that led to the publications, but they were subsequently excluded from the study based on the exclusion criteria.

**Table 2.** Search strings that were excluded by the exclusion criteria (EC).

| TS = (BPMN AND " ") | S | EC | TS = (BPMN AND " ") | S | EC |
|---|---|---|---|---|---|
| Adult | 1 | EC3 | Nerve | 1 | EC3 |
| Aerospace | 1 | EC3 | Nutrition | 1 | EC3 |
| Aesthetic | 1 | EC3 | Peripheral | 2 | EC3 + EC4 |
| Anatomy | 1 | EC3 | Plastic | 2 | EC3 |
| Biological | 3 | EC1 + 2EC3 | Preventive | 2 | EC3 |
| Disaster | 1 | EC3 | Ultrasound | 2 | EC3 |
| Forensic | 4 | EC3 | Urgent | 1 | EC3 |
| Generalist | 1 | EC3 | Vascular | 1 | EC4 |

*2.5. Synthesis of Results*

We first subjected the publications listed in the Results section to bibliometric analysis and then evaluated the studies according to their content. The bibliometric analysis describes, analyzes, and summarizes the development of the area of quantitative research of the literature. This analysis allows the summarizing of the results from the micro-level (institutes, scientists, and universities) to the macro-level (countries and continents) [42]. For science mapping and research, we used VOS Viewer, Bibliometrix, Venn diagrams, bar or bubble graphs, and various statistical methods, which are specified below [43,44].

VOSviewer is a software tool for constructing and visualizing bibliometric networks [45]. Bibliometrix is an open-source tool for quantitative research in scientometrics and bibliometrics [46].

The Venn/Euler diagram graphically represents the relationships a basic set of keywords. Euler diagrams are considered an effective means of visualizing containment, intersection, and exclusion. The goal of such graphs is to communicate scientific results visually. Leonhard Euler first popularized the principle of labeling closed curves in the paper [47]. Alternative names for Euler diagrams include "Euler circles". They can also be incorrectly called Venn diagrams. Venn diagrams require all possible curve intersections to be present, so they can be seen as a subset of Euler diagrams, that is, every Venn diagram is an Euler diagram, but not every Euler diagram is a Venn diagram. John Venn introduced Venn diagrams a hundred years after Euler in the paper [48]. A Venn diagram is a schematic graph used in logic theory to depict collections of sets and to represent their relationships.

A conceptual structure map represents a map of a scientific field by performing correspondence analysis (CA), multiple correspondence analysis (MCA), or metric multidimensional scaling (MDS), as well as clustering of a bipartite network of terms extracted from the keyword, title, or abstract field. The object of correspondence analysis (CA) is to analyze categorical/categorized data that are transformed into cross-tables and to demonstrate the results graphically. Thematic evolution analysis is based on co-word network analysis and clustering. The methodology is inspired by the proposal [49].

## 3. Results

When selecting a basic sample of documents, we focused on the WOS database, as it is a database of papers that have undergone the peer-review process. We compiled search strings according to the documents [39,40], from which we selected a total of 272 phrases listed in Table A1. From these phrases, we collected search strings and identified 485 documents, of which 108 met all the criteria. We subjected these documents to various analyses such as bibliometric analysis, content analysis, or systematic literature research, and their results are presented in this section.

Figure 2 introduces the "word cloud" based on the analysis of abstracts.

An overview of the publications and their corresponding "search queries" is given in Table 3. In this table, the reader will also find the number of documents found in the given string (S), the number of papers that were not excluded based on exclusion criteria (I), and their links to references.

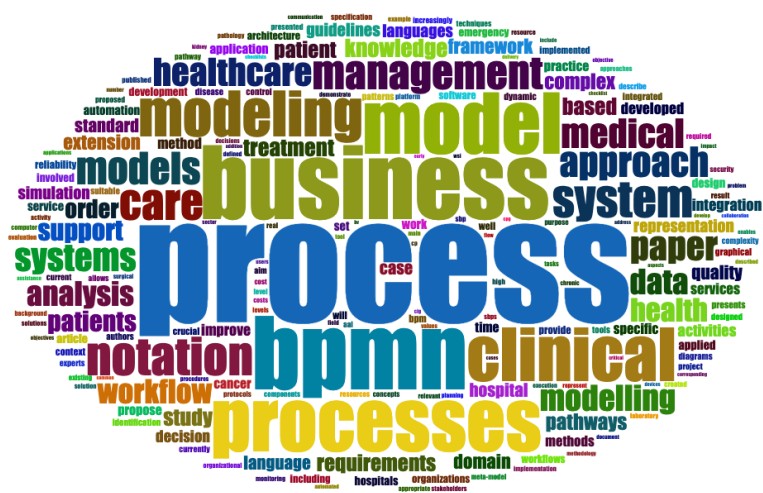

**Figure 2.** Word cloud figure based on the analysis of abstracts.

**Table 3.** Search string, number of publications found (S), and number of papers included (I) in the study.

| TS = (BPMN AND " ") | S | I | Reference |
|---|---|---|---|
| Accident | 2 | 1 | [50] |
| Adolescent | 1 | 1 | [51] |
| Behavioral | 42 | 5 | [52–56] |
| Cardiology | 1 | 1 | [57] |
| Care | 57 | 47 | [50,51,53,57–100] |
| Clinical | 51 | 42 | [57,63–67,73–79,81–83,86,87,89,91,93,99,101–120] |
| Colon | 3 | 2 | [63,106] |
| Colorectal | 1 | 1 | [63] |
| Critical care | 5 | 3 | [78,80,94] |
| Diabetes | 4 | 4 | [83,88,91,116] |
| Diagnostic | 13 | 3 | [86,106,121] |
| Disabilities | 3 | 2 | [61,122] |
| Disease | 17 | 15 | [55,59,61,64,74,75,81,83,86,88,102,111,113,121,123] |
| Emergency | 12 | 7 | [50,84,90,95,97,124,125] |
| Emergency medical services | 1 | 1 | [50] |
| Family | 14 | 1 | [126] |
| Health | 55 | 40 | [51,52,59,60,62,64,68,74,80,82,83,86,88,90–94,96,98,99,102–104, 109,111,112,115,124,127–137] |
| Health informatics | 3 | 3 | [93,103,128] |
| Hospital emergency | 3 | 3 | [95,97,125] |
| Hospital medicine | 2 | 2 | [106,109] |
| Child | 4 | 1 | [51] |
| Infectious | 1 | 1 | [102] |
| Infectious disease | 1 | 1 | [102] |
| Intensive care | 4 | 4 | [65,66,86,105] |
| Internal medicine | 1 | 1 | [138] |
| Kidney | 3 | 3 | [55,92,139] |
| Kidney diseases | 1 | 1 | [55] |
| Laboratory | 21 | 7 | [71,95,102,140–143] |

**Table 3.** *Cont.*

| TS = (BPMN AND " ") | S | I | Reference |
|---|---|---|---|
| Medical | 50 | 46 | [50,56,58,63,64,66,72,74,75,78,79,81–83,85–87,96,99,101,103,106, 108–111,113–115,120,126,128,133,134,144–155] |
| Medical research | 5 | 5 | [103,108,128,147,150] |
| Medicine | 4 | 4 | [106,109,128,138] |
| Neurosurgery | 1 | 1 | [146] |
| Occupational | 2 | 1 | [121] |
| Oncology | 2 | 1 | [63] |
| Pain | 1 | 1 | [74] |
| Pain management | 1 | 1 | [74] |
| Pathology | 5 | 4 | [122,127,128,141] |
| Physical | 42 | 5 | [52,72,82,156,157] |
| Public Health | 7 | 3 | [59,82,92] |
| Rectal | 2 | 2 | [63,106] |
| Rehabilitation | 2 | 1 | [106] |
| Spine | 1 | 1 | [122] |
| Surgery | 2 | 2 | [67,145] |
| Surgical | 6 | 5 | [101,127,141,145,146] |
| Therapeutic | 1 | 1 | [86] |

### 3.1. Main Information about the Collection

Within the searched interval of 2004–2020, we identified relevant articles only for the time interval of 2007–2020. We interpret the absence of publication before 2007 as being due to the fact that it was not until 2006 that the OMG issued the official specification for the BPMN version 1.0. In total, we identified 94 sources for 108 documents and 2182 references. On average, each record has 4.92 citations, and the average citation rate per year is 0.73.

In the introduction, we divided the analyzed sample of publications according to their type. We processed the data into a Venn diagram in the Figure 3 because there was an overlap of types. The resulting values correspond to the essence of the WOS database. The number of publications from conference outputs still predominates, but this predominance is less than 2:1. This figure represents a Venn diagram of the corresponding groups and their intersections: Documents belong to three groups or a combination thereof. These are Articles, Proceedings Papers, and Book Chapters. Most documents belong to the Proceedings Paper group (70), and the second place went to documents marked as Articles (33). The other two groups are combinations, i.e., subsets of Articles-Proceedings Papers with four papers and Articles-Book Chapters with one document.

In total, we identified 102 keywords labeled as "Keywords Plus" and 359 keywords selected by the authors and labeled as "Authors' Keywords". From the bibliometric analysis, we identified a total of 392 authors and one publication with a single author. On average, 0.276 articles are published per author, 3.63 authors per paper, and 4.76 co-authors per document. The collaboration index then comes out as 3.65.

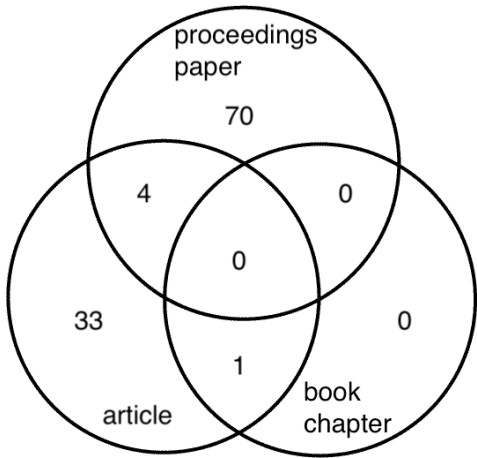

**Figure 3.** Venn diagram of distribution by document type.

*3.2. Article Metrics*

The meta-analysis of articles begins with measuring the annual production of publications. We expanded these simple data by dividing the publications by research area. We also marked the milestones of the BPMN updates in the graph.

Figure 4 shows the progression and trends of annual scientific production of scholarly publications from 2007 to 2020 with BPMN version milestones. The year 2007 is discussed above and corresponds to the statement of the official BPMN 1.0 by the OMG. The upgrade from BPMN version 1.2 to 2.0 meant greater user friendliness and higher usability. This statement is evident from the slight increase in the number of BPMN concepts published in 2012. The most recent version, 2.0.2, which is, in the authors' opinion, the most user friendly—even although it also has its mistakes—has brought a new wave of publications since 2015.

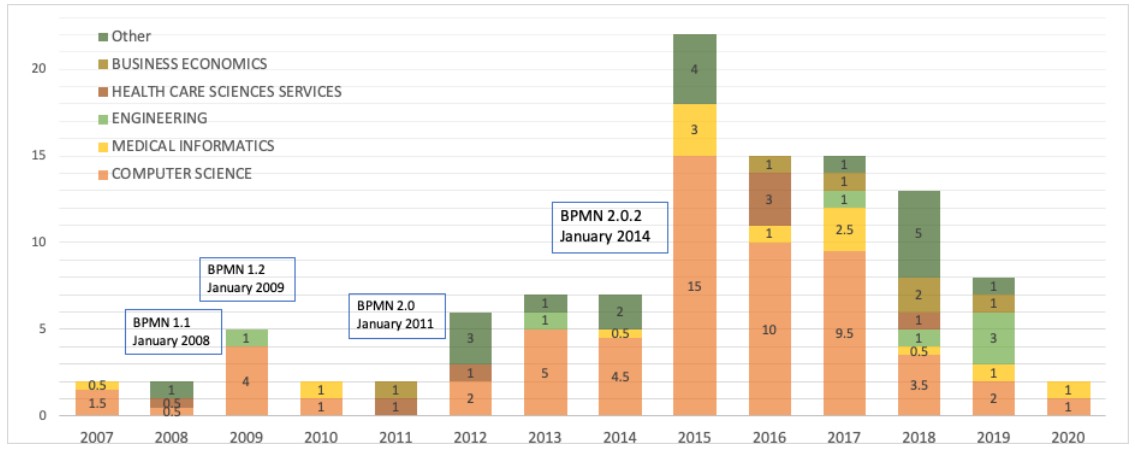

**Figure 4.** The publication trends with the BPMN version updates; authors' own processing.

At the same time, we divided the documents according to the research areas defined by Clarivate Analytics in Figure 5. Each article can belong to more than one research area. In the text, we also present both values: total and (relative affiliation). The largest group consists of publications included in the research area of computer science. In total, we identified 64 (59.5) papers in this area. Eighteen documents belong to the "Other" group, and fifteen (11) belong to the Medical Informatics group. Seven publications belong to the group of Engineering and Health Care Sciences Services (6.5). The last group, Business Economics, contains six documents.

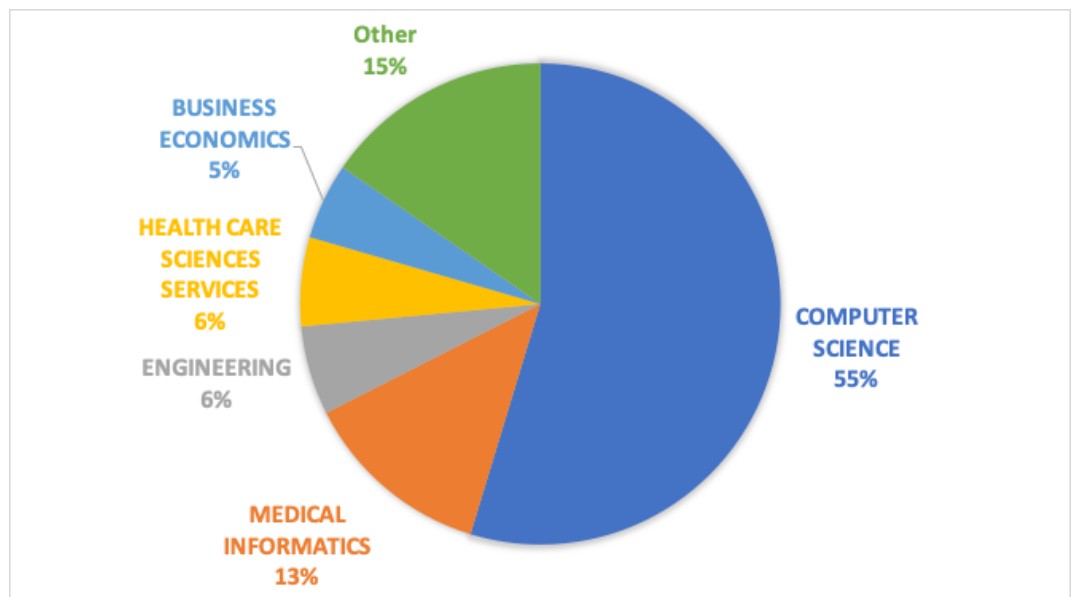

**Figure 5.** Research areas; authors' own processing.

### 3.3. Author Analysis

We analyzed the authors according to publication production over time and total citations per year, and present a graphical overview in Figure 6. The orange line represents the authors' active years; the circles represent the number of publications published in a given year and their total citations. If we look at the pure data of the documents, we can conclude that, on average, the authors published their work in 2016. The highest year of publication is 2020; the minimum is 2013 (externalities are Rodriguez et al. [70] for 2007 as well as Garcia Rojo et al. [127] and Rolon et al. [133] for 2008). The lower quartile is 2015, and the upper quartile is 2017.

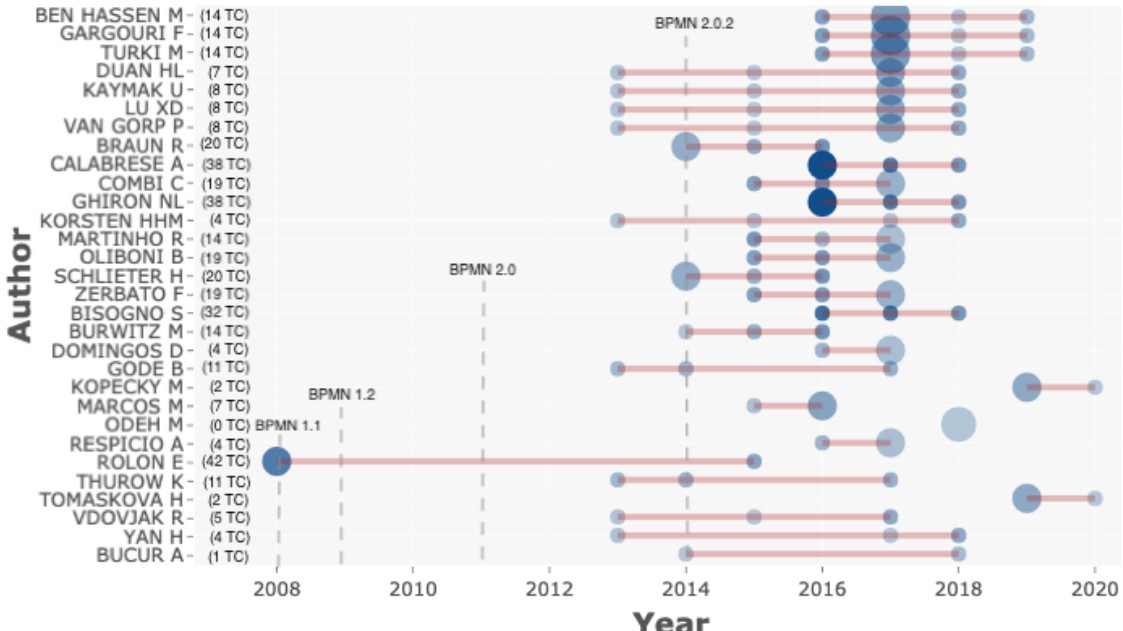

**Figure 6.** Top 30 authors' production over time with marked BPMN milestones. The total number of citations by the author is given in parentheses.

However, if we take the 30 best authors of the given sample, apart from a single author, they all started publishing only after 2013, i.e., only with the BPMN version 2.0. This single author

(Rolon et al. [133]) is one of the externalities, as mentioned above. Another externality (Garcia Rojo et al. [127]) then became an essential and cited source for younger publications. We report the historical development of citations for 20 records in Figure 7.

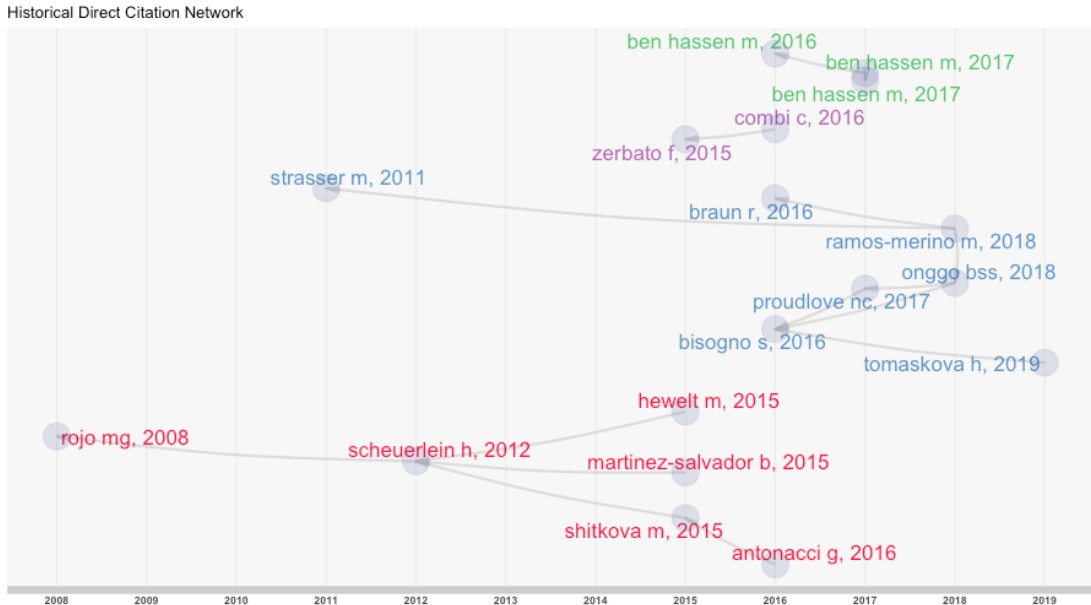

**Figure 7.** Historical direct citation network.

The average value of total citations is three; the median is equal to two. The lower quartile is on the border of zero citations, and the upper quartile consists of five citations. The two externalities correspond to sixteen citations in [95] and 24 citations in [127].

Figure 8 represents the scientometric analysis according to Lotka's law. Lotka's law describes the frequency of publication by authors in a given field. In our study, 83.7% of the authors published one publication, and 8.9% of the authors published two publications. Three documents were published by 3.3% of the authors, and 2.3% published four papers, etc.; a maximum of seven publications were published by 0.76% of the authors. The dashed line shows the theoretical distribution according to Lotka's law.

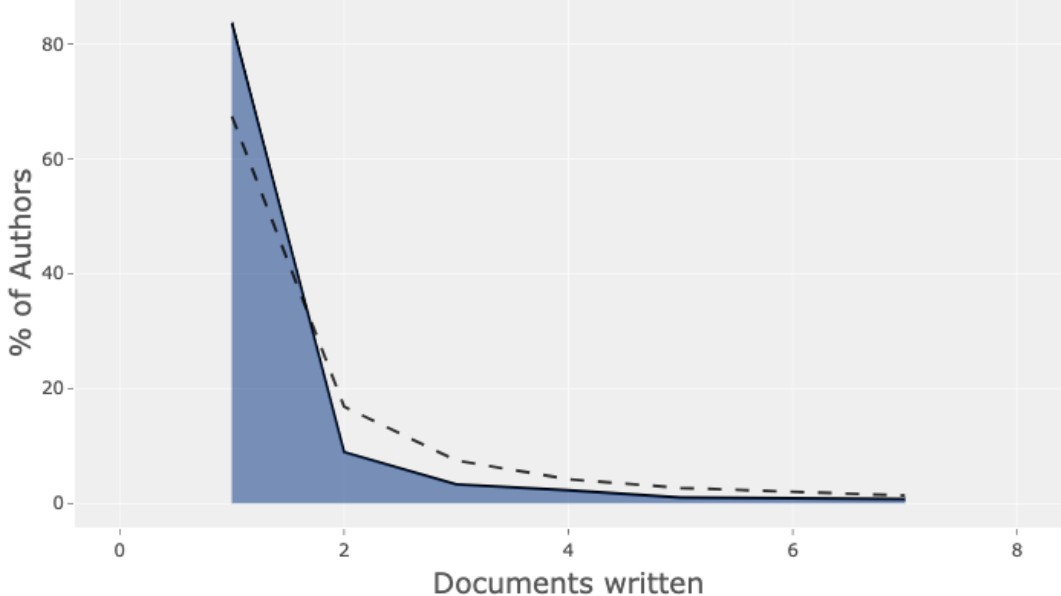

**Figure 8.** The frequency distribution of scientific production.

### 3.4. Country and Affiliation Metrics

The authors of the selected publications are based on almost every continent of the world, as can be seen in Figure 9. In the following paragraph, we list the individual countries and their numbers of documents: 40 documents in Germany, 21 documents in Spain, 19 documents in the UK, 18 documents in Italy and Tunisia, 16 documents in the Netherlands, 13 documents in Brazil, 11 documents in China, 9 documents in Austria and the USA, 8 documents in Jordan and Serbia, 7 documents in Finland, 6 documents in Mexico, 5 documents in the Czech Republic, France, and Romania, 3 documents in Georgia and Russia, 2 documents in Belgium, Canada, Denmark, and Morocco, and 1 document in Chile, Colombia, Ghana, Greece, Hungary, Israel, Poland, and Sweden.

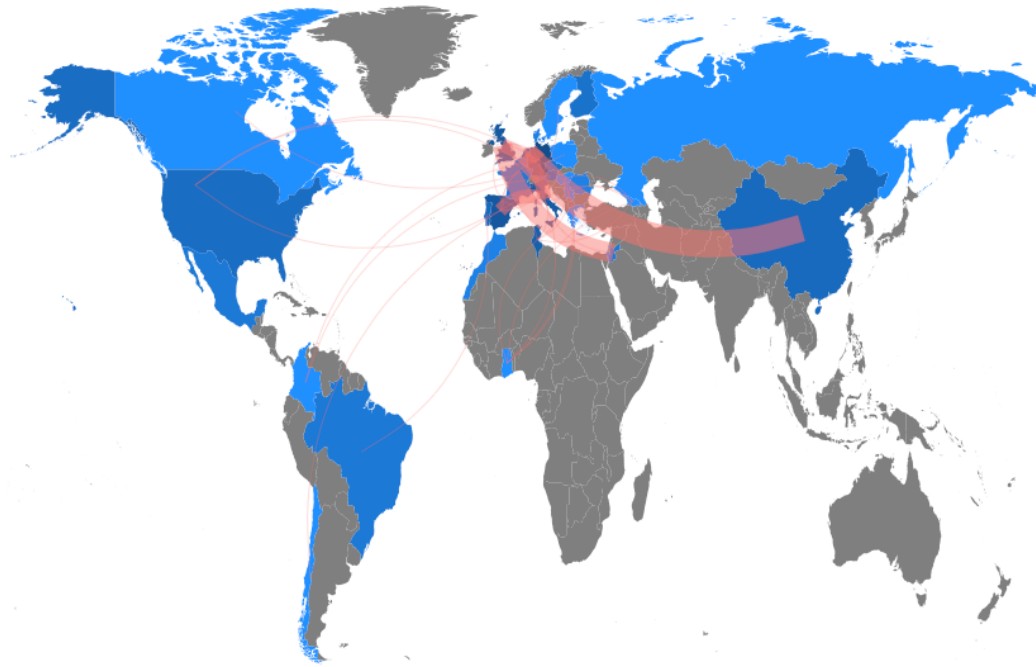

**Figure 9.** Collaboration world map.

Figure 10 contains the list of countries in the data frame; the countries of the first authors of the article were used to make this table. The figure also includes the frequency of publication and the distribution of papers by an author(s) affiliated with a single country or by an author(s) affiliated with multiple countries (at least one author from the country in the list published with an author(s) from another country (or countries)).

We show the cooperation of particular countries in Figure 11. Connections with a multiplicity higher than one are marked with the appropriate value. Unmarked links have a value of one.

We also analyzed countries by citation and plotted the results in Figure 12. Surprisingly, Chile occupied the first position, but this is because a single article [70] received 105 citations. That is the most cited of all documents. Unfortunately, this document has its citations outside the selected papers, and so it was not listed in Figure 7.

We continued the analysis in greater depth with institutions, and we analyzed these affiliations with the Keywords Plus. We show the resulting layout of the Keywords Plus in Table 4. This table lists Keywords Plus depending on the authors' affiliations and the number of relevant documents. Due to the large number of combinations, these values are the most numerous.

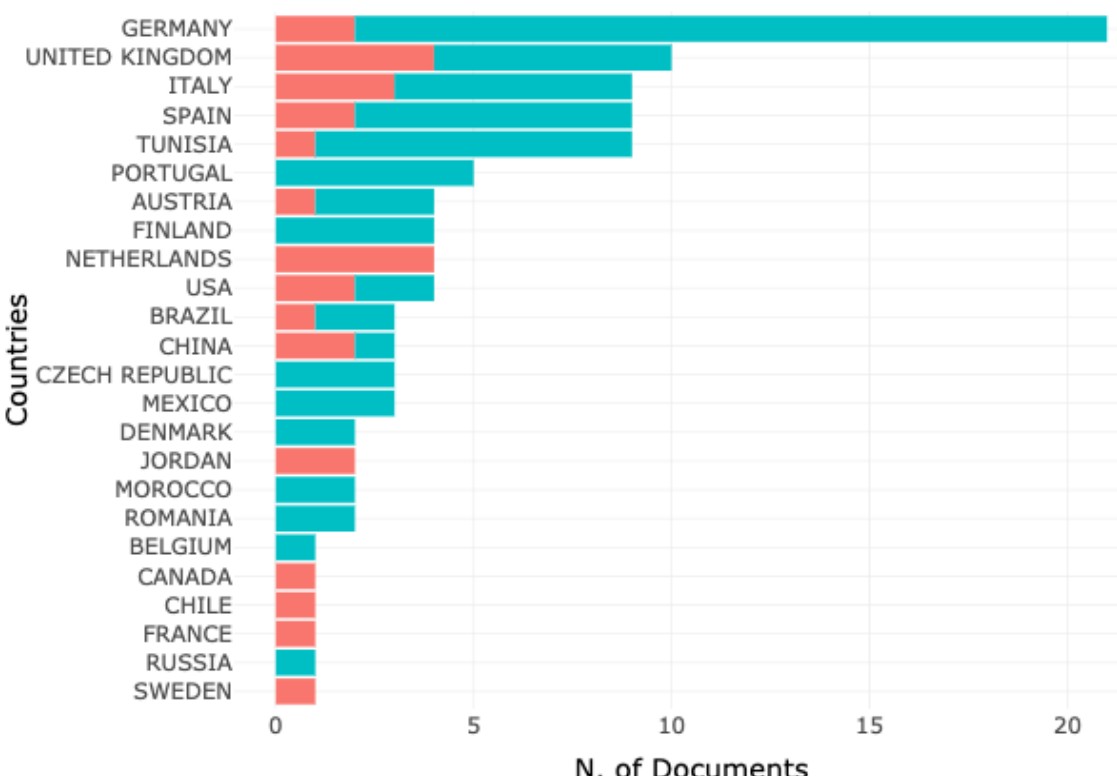

**Figure 10.** Corresponding authors by country (green = single-country publications; red = multiple-country publications).

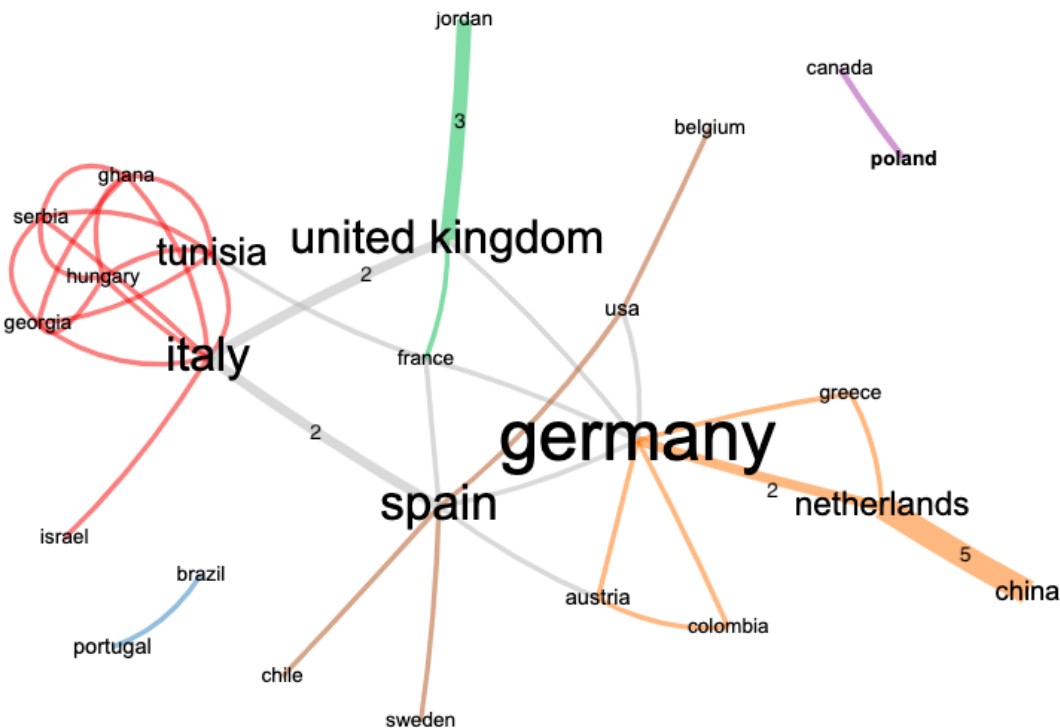

**Figure 11.** Country collaboration map (numbers of connections greater than one are marked by the corresponding value).

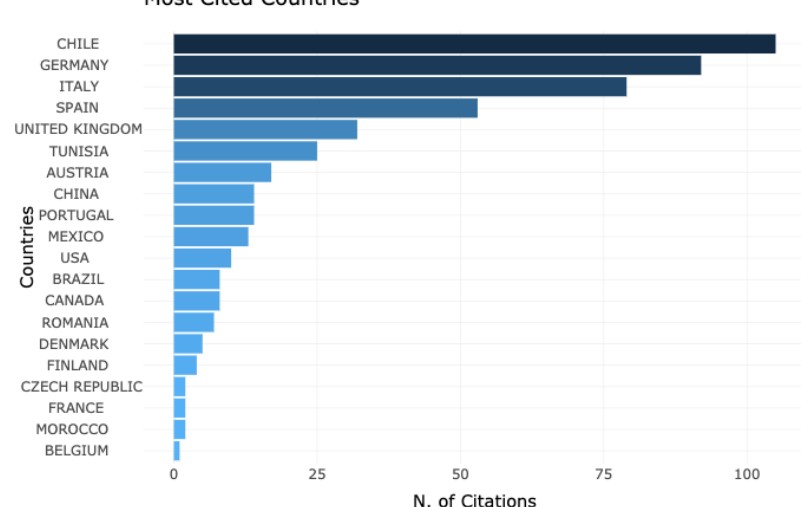

**Figure 12.** Most cited countries.

*3.5. Keywords*

All 108 documents were subjected to two keyword analyses. Both cases were analyses under the following conditions: all keywords (authors' keywords or Keywords Plus), full counting, and the minimum number of occurrences of the keyword was one. When analyzing keywords, we identified a total of 334 authors' keywords and 102 Keywords Plus. While the authors' teams choose common keywords, Keywords Plus are index labels that are automatically generated from the titles of cited articles. Keywords Plus phrases must appear more than once in the bibliography and are ordered from multiword phrases to single terms [158].

We analyzed both types of keywords and plotted the co-occurrence results in Figures 13 and 14. Both images represent a network of keyword combinations. Their color corresponds to the average year of their use in publications. A total of 90% of the authors' keywords are all linked, which can be seen in Figure 13. There are 300 linked keywords centering around the term "BPMN". We analyze the authors' keywords in the next section according to historical development.

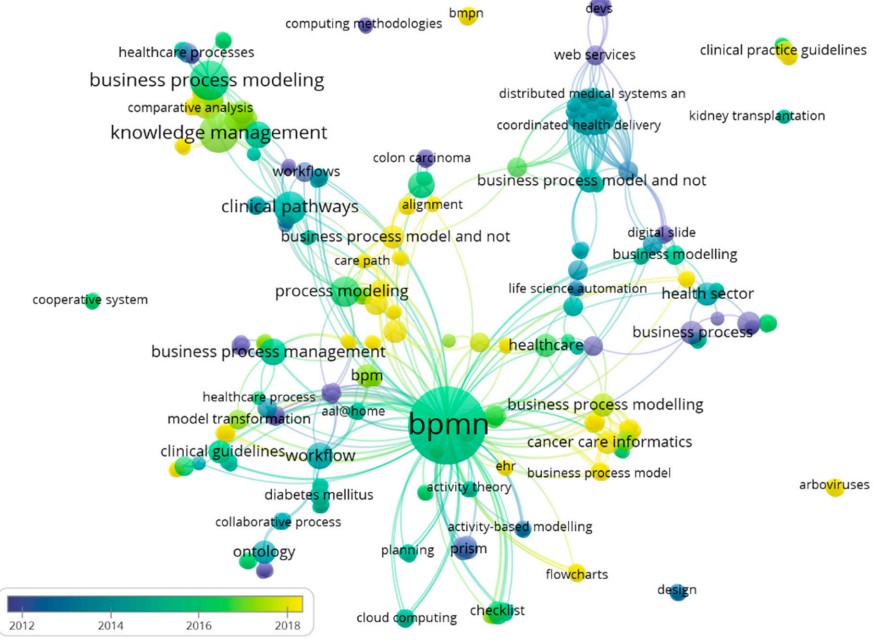

**Figure 13.** Co-occurrence of authors' keywords.

**Table 4.** Keywords Plus according to affiliation.

| | Univ Birmingham | Univ Hosp Essen | Univ Sao Paolo | Univ Sfax | Univ Murcia | Univ Verona | Philips Res | Zhejiang Univ | Eindhoven Univ Technol | Univ Saarland | Tech Univ Dresden | Univ Roma Tor Vergata | Sch Technol and Management | Univ Lancaster | Univ Manchester | Univ Lisabon | Univ Hradec Kralove |
|---|---|---|---|---|---|---|---|---|---|---|---|---|---|---|---|---|---|
| semantics | 2 | | | | | | | | | | | | | | | | |
| trials | | 4 | | | | | | | | | | | | | | | |
| extending BPMN | | | 2 | | | | | | | | | | | | | | |
| pathways | | 4 | | | 1 | 1 | | | | | 1 | | | | | | |
| support | | 4 | | | 1 | 1 | | | | | | 1 | | | | | |
| process models | | | | | | 2 | | | | | | | | | | | |
| challenges | 2 | | | | 1 | 1 | 1 | | | 1 | | 1 | | | | | |
| BPMN | 2 | | | 1 | | | 1 | | 2 | 2 | | | | | | | |
| systems | | 4 | 3 | | 1 | 1 | | | | | | | 1 | | | | |
| practical gudelines | | | | | 1 | 1 | 1 | 1 | 1 | | | | | | | | |
| care | | 4 | | | 1 | 1 | | | | | | 1 | | | | | 2 |
| guidelines | | | | | 1 | 1 | 2 | 1 | 1 | 2 | 3 | | | | | | |
| decision support | | | | | | | | 1 | 1 | | | | | | | | |
| management | | | | 1 | | 2 | | | | | | | 1 | | | | 1 |
| system | | | | | | | | 1 | 1 | | | | | | | | 1 |
| implementation | | | | | | | | 1 | 1 | | 1 | 2 | | 2 | 2 | | |
| health-care | | | 3 | | | 2 | | | | | | | 1 | | | 5 | |
| information systems | | | | | | | | | | | 3 | | | | | | |
| framework | | | | | | | | | | | | 3 | | 2 | 2 | | |
| reality | | | | | | | | | | | | 2 | | 2 | 2 | | |
| quality | | | | | | | | | | | | | 1 | | | 2 | |
| dementia | | | | | | | | | | | | | | | | | 3 |
| population | | | | | | | | | | | | | | | | | 1 |
| survival | | | | | | | | | | | | | | | | | 1 |
| deficits | | | | | | | | | | | | | | | | | 1 |
| population balance model | | | | | | | | | | | | | | | | | 2 |
| economic evaluation | | | | | | | | | | | | | | | | | 2 |

The keywords plus are only 78% linked, one central term is missing, and the structure has a large number of separate externalities. Although BPMN is one of the most commonly used terms, its network is comparable to such terms as "Implementation", "Guidelines", "Management", "Pathways", or "Framework".

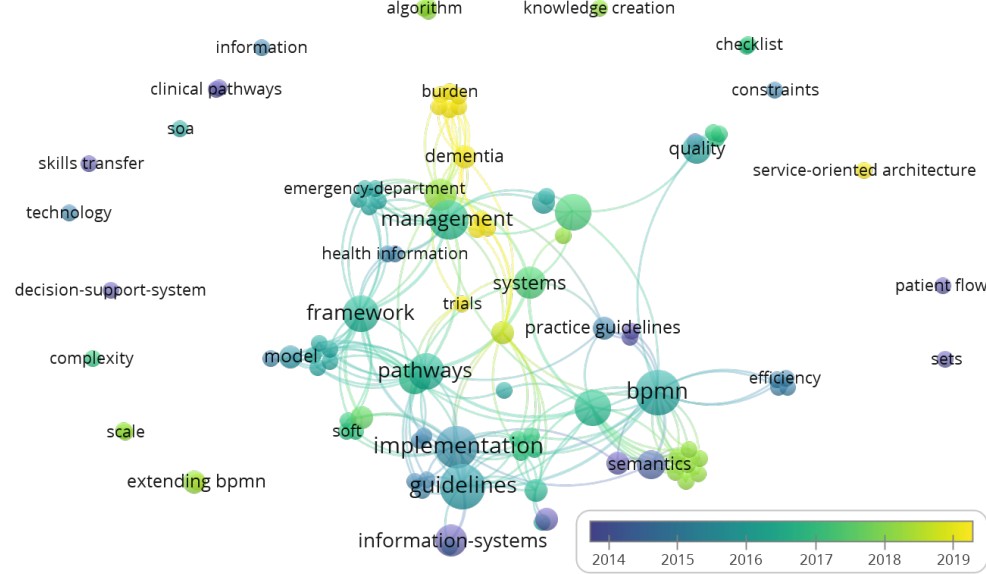

**Figure 14.** Co-occurrence of Keywords Plus.

### 3.6. Thematic Evolution and Topic Trends

The last part was devoted to advanced analyses, where we focused on the thematic evolution of some key terms, cluster analysis, or analysis of the purposes of documents.

In the following figure, Figure 15, we have created an analysis of the development of the authors' keywords. Thematic evolution analysis is based on co-word network analysis and clustering [49]. We analyzed the maximum, i.e., 300 words, and chose the inclusion index weighted by word occurrence as the weighting index. We decided to use four cutting years—2013, 2015, 2016, and 2018. The year 2013 corresponds to the average time of the first publications by the authors. In 2015, there was a global maximum of published documents. The year 2016 is the average year of publication. In 2018, papers were published in all research areas.

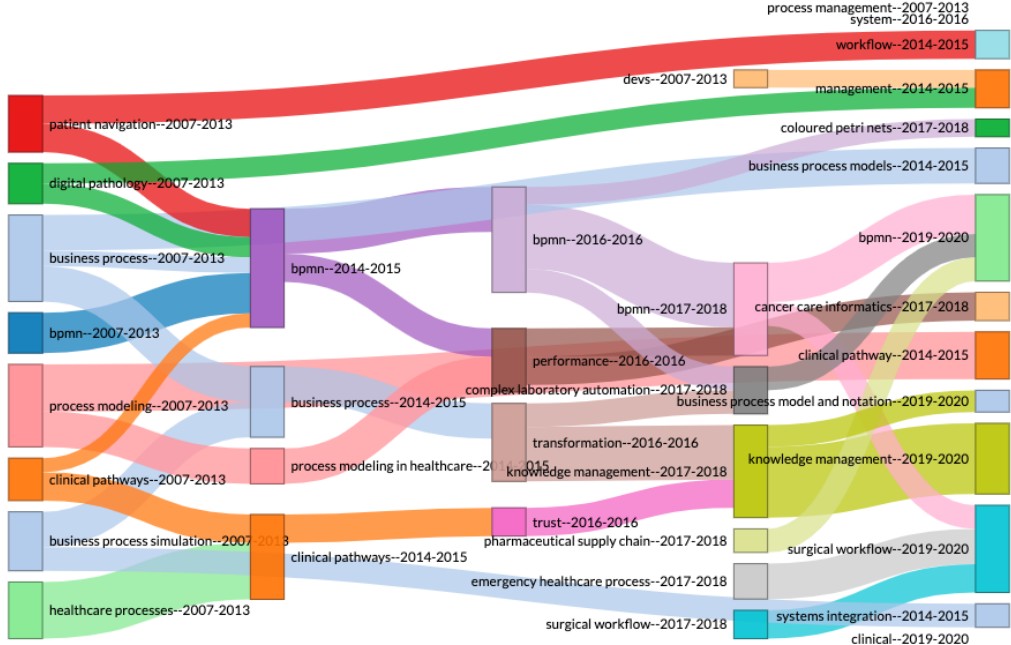

**Figure 15.** Thematic evolution of authors' keywords.

In the following section, we list the names of clusters for each period, for which we have compiled a thematic map. The thematic map is based on co-word network analysis and clustering.

The first period contains the following 12 clusters, which are shown in Figure 16.

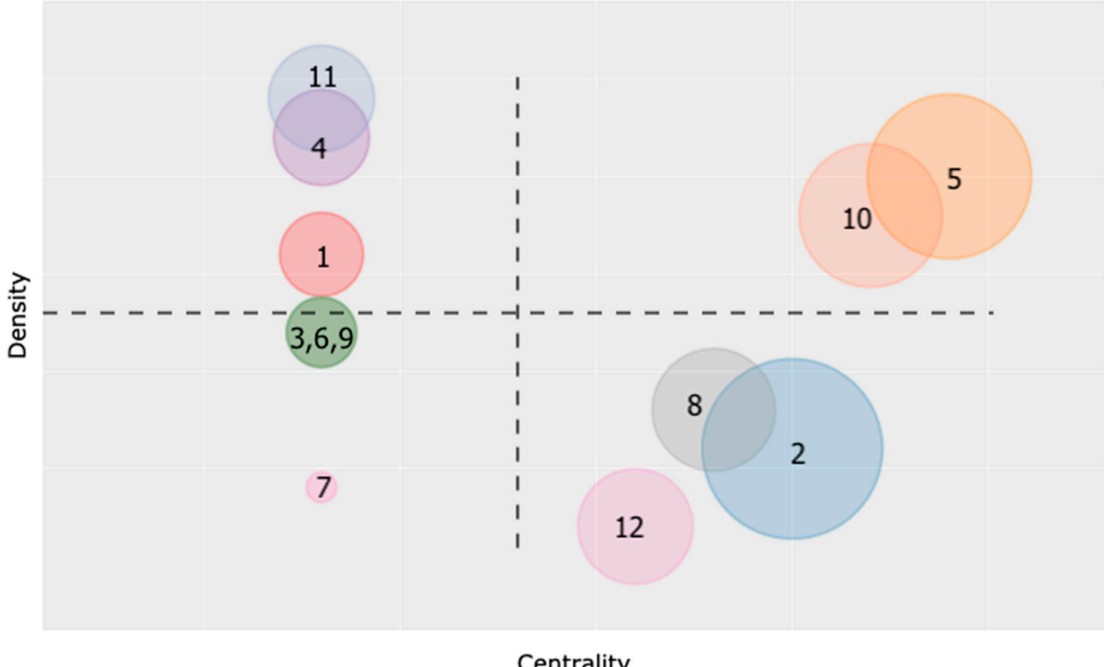

**Figure 16.** Time Slice 1 (2007–2013). 1. Business process simulation; 2. BPMN; 3. haptic simulator; 4. digital pathology; 5. process modeling; 6. patient navigation; 7. medical informatics; 8. clinical pathways; 9. healthcare processes; 10. process management; 11. devs; 12. business process.

The second period contains the nine clusters shown in Figure 17.

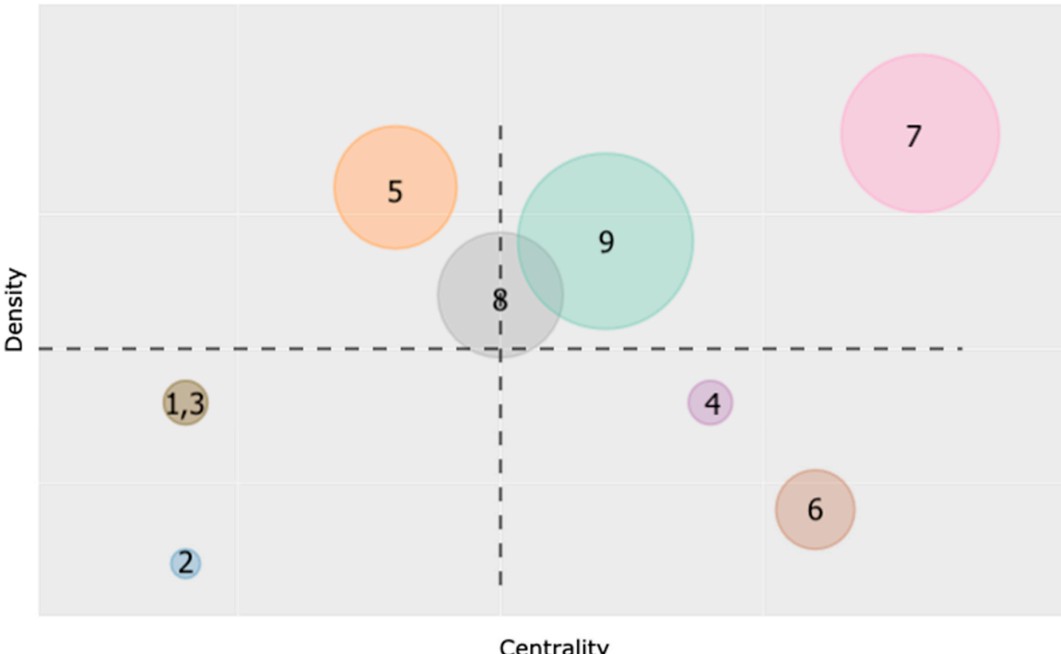

**Figure 17.** Time Slice 2 (2013–2015). 1. Business process models; 2. clinical pathway; 3. process modeling in healthcare; 4. business process; 5. clinical pathways; 6. systems integration; 7. management; 8. workflow; 9. BPMN.

The third period contains the six clusters shown in Figure 18.

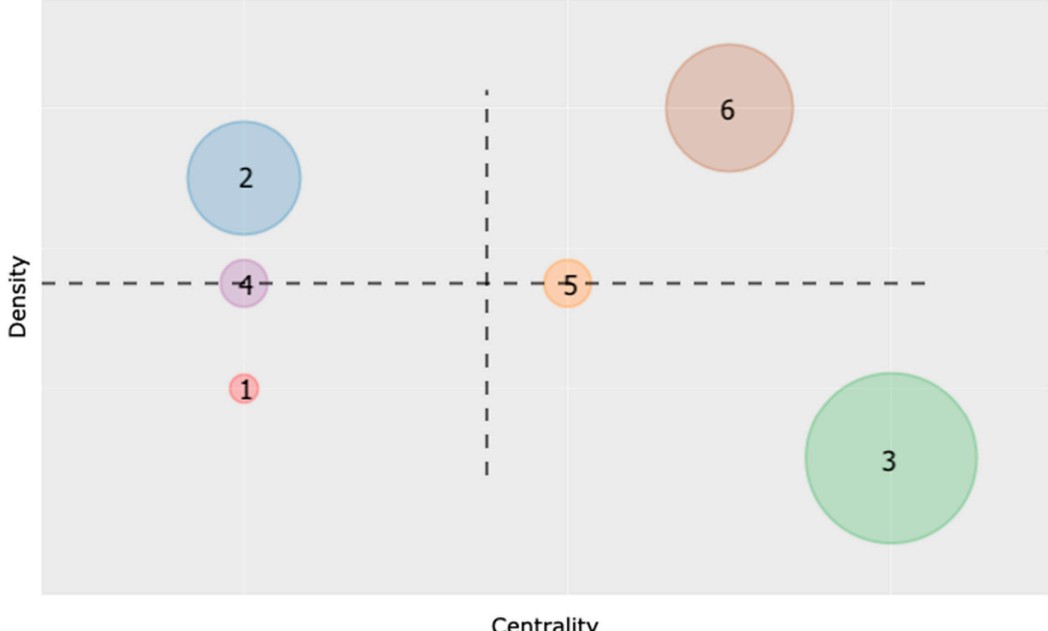

**Figure 18.** Time Slice 3 (2015–2016). 1. Clinical practice guidelines; 2. system; 3. BPMN; 4. trust; 5. performance; 6. transformation.

The fourth period contains the 11 clusters shown in Figure 19.

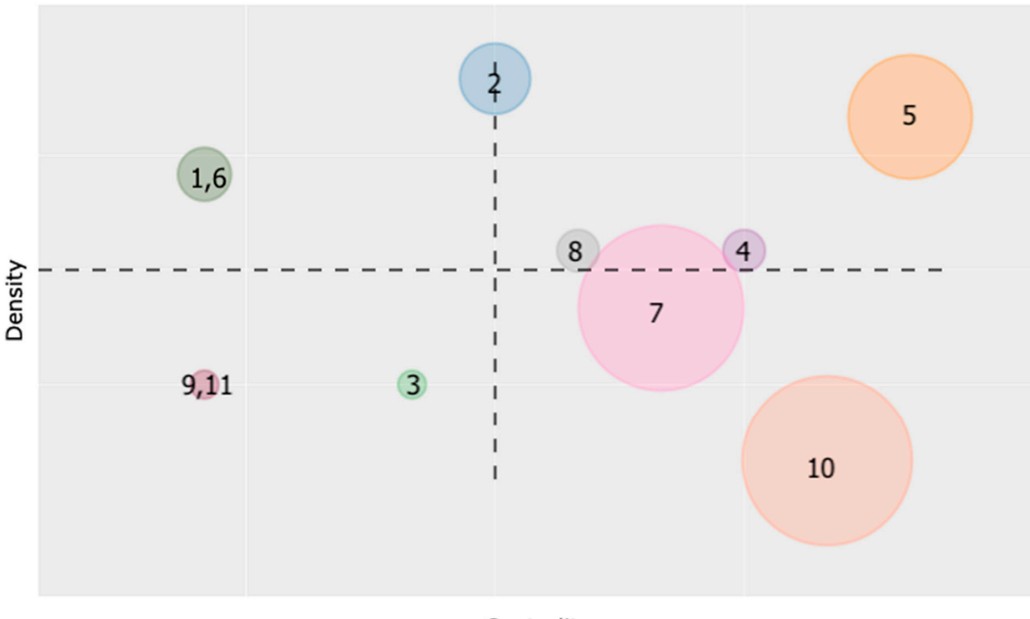

**Figure 19.** Time Slice 4 (2016–2018). 1. Clinical decision support; 2. colored petri nets; 3. clinical governance; 4. business process model; 5. cancer care informatics; 6. pharmaceutical supply chain; 7. BPMN; 8. surgical workflow; 9. complex laboratory automation; 10. knowledge management; 11. emergency healthcare process.

The fifth period contains the six clusters shown in Figure 20.

For the next analysis, we chose the topic trend analysis, where we focused on the titles of publications. We used words in the investigation that occurred at least twice and a maximum of

seven terms per year. We processed the results in Figure 21, and verbally describe them in the following paragraph.

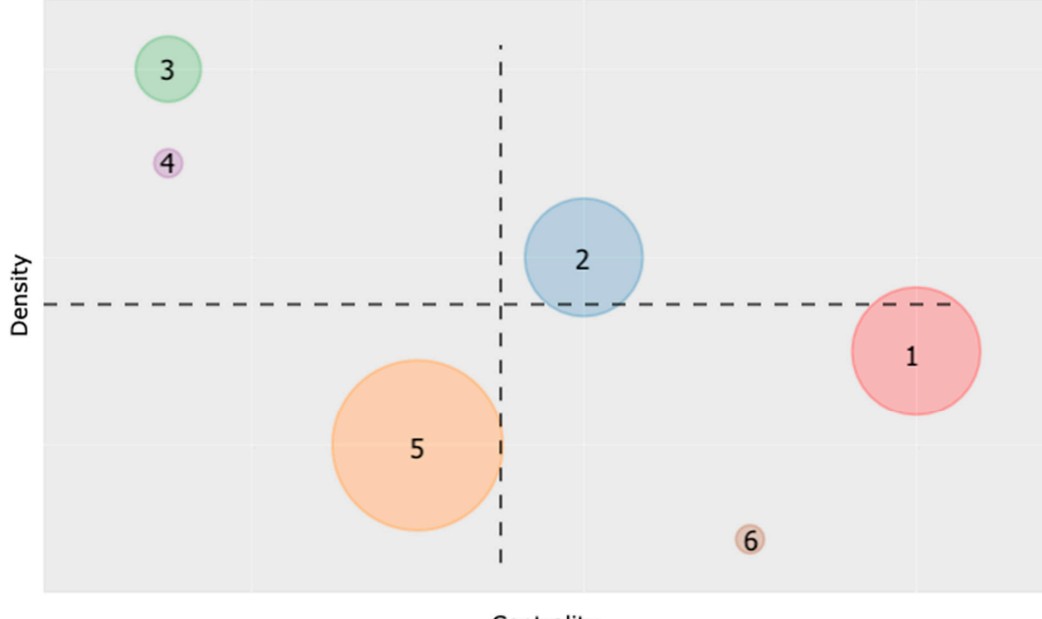

**Figure 20.** Time Slice 5 (2018–2020). 1. Business Process Model and Notation; 2. clinical; 3. knowledge management; 4. arboviruses; 5. BPMN; 6. surgical workflow.

**Figure 21.** Trending topics for titles.

In the following section, we present individual terms and their occurrence according to the relevant years.

- 2010—Emergency with four occurrences and PATHOLOGY with two occurrences.
- 2012—SIMULATIONS with three occurrences, REQUIREMENTS with three occurrences, APPROACHES with two occurrences, COMPLIANCE with two occurrences, SECTOR with two occurrences, RECORDS with two occurrences, and QUANTITATIVE with two occurrences.
- 2013—PROBABILISTIC with three occurrences.
- 2014—HEALTH with 11 occurrences, MODELING with 11 occurrences, DESIGN with five occurrences, WORKFLOWS with four occurrences, AUTOMATION with four occurrences, MOBILE with three occurrences, and DEVELOPMENT with three occurrences.
- 2015—PROCESS with 36 occurrences, BPMN with 30 occurrences, CLINICAL with 23 occurrences, MODEL with 10 occurrences, INTEGRATED with six occurrences, DATA with four occurrences, and TREATMENT with four occurrences.
- 2016—BUSINESS with 28 occurrences, MODELING with 21 occurrences, PROCESSES with 19 occurrences, CARE with 14 occurrences, PATHWAYS with 12 occurrences, HEALTHCARE with 10 occurrences, APPROACH with nine occurrences, ANALYSIS with nine occurrences, and FRAMEWORK with nine occurrences.
- 2017—MANAGEMENT with 12 occurrences, SENSITIVE with seven occurrences, KNOWLEDGE with seven occurrences, CASE with seven occurrences, DYNAMIC with four occurrences, EXTENDING with three occurrences, META-MODEL with three occurrences, LABORATORY with three occurrences, and RELIABILITY with three occurrences.
- 2018—REPRESENTATION with seven occurrences, BASED with six occurrences, CANCER with five occurrences, HOSPITAL with three occurrences, OMG with two occurrences, STANDARDS with two occurrences, CMMN with two occurrences, DMN with two occurrences, COMPARATIVE with two occurrences, EVENT with two occurrences, TASK-TIME with two occurrences, MATRIX with two occurrences, VARIANCE with two occurrences, and THERAPY with two occurrences.
- 2019—PATIENTS with two occurrences.
- 2020—ALZHEIMER'S with two occurrences and DISEASE with two occurrences.

From the analysis, we can conclude that until 2012, when the authors already made full use of the potential of BPMN 2.0, the topics only concerned the description of specific processes (Emergency and Pathology). Since 2012, there has been a significant development and expansion of topics from simulation, through mobile technologies, to dynamic decision-making. Since 2018, the subjects have focused on the specialization of the area and the specification of processes.

*3.7. Purpose Analysis*

Finally, we performed a purpose analysis, thanks to which we identified the following objectives. Healthcare processes:

- Direct:

  - Clinical Guidelines (CG) = Safety Checklist and Care Plan [55,57,63–66,73–75,78,79,83,87,105–108,111,113,118–120];
  - Clinical processes (CP) = Care Path [55,57,63–65,67,73–78,81–83,86,87,89,93,99,101–120].

- Indirect:

  - Legal [109,130];
  - Administrative [106,115,127];
  - Financial = Cost and Care costs [51,59,67,74,81,83,86,99,114,121,123,128,140,145,156];

–   Regulatory [130,138].

Health Information Technology:

- Electronic Health Records (ERH) [58,64,82,91,111,115,136];
- Personal Health Records (PHR) [58,74];
- Medical Practice Management software (MPM) [75,134];
- Health Information Exchange (HIE) [51,52,58–60,62,64,68,70,74,80,82,83,86,88,90–94,96,98,99,102–104,109,111,112,115,124,125,127–137];
- Ambient-Assisted Living (AAL) [62,98,156];
- Decision  (DML)  [50,54,55,57,60,63,64,66,68,74,78,80,83,86,94,97,105,111,113,121,122,124,126,133,134,138,146,152–154];
- Internet of Things (IoT) [54,63,64].

Reason for care:

- Alzheimer's Disease (AD) [59,81,123];
- Arbovirus infection [102];
- Cancer [63,69,71–74,83,105,106];
- Contraceptive [64];
- Diabetes Mellitus [83,88];
- Elderly [61]
- Chronic Obstructive Pulmonary Disease [86,111];
- Kidney disease [55];
- Occupational diseases [121];
- West Nile Virus [102].

*3.8. Research Questions*

At the beginning of the study, we identified three research questions. We will now summarize the answers to these questions based on the results of the analysis above.

3.8.1. RQ1: Have the Different Versions of the BPMN Brought about a Change in Its Use in the Medical Domain?

The effect of increasing user friendliness and expanding graphic elements is evident, for example, in Figures 4, 15, and 21. Whether it was an increase in the number of published publications or the expansion of the BPMN's applications, we can consider RQ1 to be confirmed.

3.8.2. RQ2: In Which Medical Specialization Is the BPMN Used?

We answered this research question at the beginning of the study when most of the searched queries could not assign publications in the WOS database. As can be seen in Table A1, there are still many medical specializations where use of the BPMN has not yet been published. RQ2 is not confirmed.

3.8.3. RQ3: What Type of Use Is Made of the BPMN in These Specialities?

The analysis of the purpose and, thus, of the application was part of the final purpose analysis.   Here, we can say that most, i.e., 37%, of the publications focused on clinical processes. "Health Information Exchange" came in second and "Decision-Making" came in third. RQ3 was confirmed.

## 4. Limitations of the Study

There are three major limitations in this study that could be addressed in future research. These limitations are listed and discussed in the following section.  The study was not limited in

terms of access to data, as our research does not contain any sensitive or personal data. Furthermore, there were no conflicts during the analysis resulting from cultural bias or other personal issues.

Despite the limitations listed below, we believe that this is a useful study, bringing a new perspective. Furthermore, we believe that our paper shows many research gaps and found some opportunities for future research.

*4.1. Sample and Selection*

The first limitation of our study is the selection of a sample for the literature review. We analyzed papers from a single scientific database only, so the analyzed sample may not reflect the general population. It is thus possible that the selection of publications published in the WOS database can be described as having "selection bias". The WOS database was chosen as a guarantor of the quality of publications and as the most well-known field-wide database. Of course, the WOS database is not the only suitable source of evidence for systematic search. However, for example, the Google Scholar database contains many materials that have not been verified by any review process. For this publication, however, we decided to focus only on the WOS database because it is known and sought after in most scientific fields. Of course, we plan further research, which will contain more databases, but it will not focus on the analysis of medical specializations, but, in general, on health care.

*4.2. Methods Used to Collect the Data*

Another limitation of the study is the way we collected the data. We focused on specific medical specializations selected from the two official documents, and thus several publications focusing on general health and health care could be excluded from the study. Our method was also time-consuming and technically demanding in the data collection phase. In further research, we plan the opposite procedure, i.e., from general queries to specialization.

*4.3. Time Constraints*

We limited the study to the period from 1 January 2004 to 10 June 2020. The year 2004 was chosen as the starting point, as it was when the Business Process Management Initiative (BPMI) created the BPMN. However, we discussed two options for the upper limitations of the study. In the first variant, if we would choose the end of 2019, we would deprive the literature review of current and beneficial publications published in 2020. However, the annual values are complete. The second variant limited the study to the date on which it was still possible to process the results before the submission deadline. This variant brought current but incomplete results from 2020. We chose the second variant, and so the values of almost half of the year 2020 are in the study.

## 5. Conclusions

The article presents a systematic literature search, which focused on the connection of the BPMN and medical specializations in publications listed in the Web of Science database by Clarivate Analytics. We established the basic search queries based on two official international medical documents. For these 272 basic terms, we identified 485 papers in the database. We subsequently subjected them to a systematic review according to our exclusion criteria. We further analyzed the resulting 108 articles using bibliometric analyses or advanced content analysis.

Due to the broad scope of the medical field, it was no surprise that for almost 85% of the sought-after medical specializations, we could not identify any publications in the given database that included the BPMN. However, the influence of this graphic language is significant. Its development and, thus, the improvement of user friendliness is evident, for example, in increasing publishing activity after the advent of the "better version" or by expanding application possibilities. The article also shows the results of analyses of types of publications, as well as analyses of authors, their affiliations with countries or institutions, and international cooperation on publications. The keyword analysis showed a diametrical difference between the authors' keywords and the so-called Keywords Plus.

The difference is noticeable not only in their dispersion and volume, but mainly in their co-occurrence. Finally, we analyzed the publications for the development of the authors' keywords, the trends in publication titles, or the main goals.

**Author Contributions:** Conceptualization, H.T. and M.K.; methodology, H.T.; validation, H.T.; formal analysis, H.T.; investigation, H.T. and M.K.; writing—original draft preparation, H.T.; writing—review and editing, H.T. and M.K.; visualization, H.T.; supervision, H.T. All authors have read and agreed to the published version of the manuscript.

**Funding:** The research was supported by a GACR 18-01246S and by the Faculty of Informatics and Management UHK Specific Research Project.

**Conflicts of Interest:** The authors declare no conflict of interest.

## Appendix A

**Table A1.** Term list—phrases are divided into No/Yes groups depending on if the included publications exist.

| No | Yes |
|---|---|
| Aerospace; Aesthetic; Aesthetic surgery; Allergology; Allergy and immunology; Anaesthesia; Anaesthesiology; Anaesthetics; Anatomy; Anesthesia; Anesthesiology; Ankle surgery; Arthritis; Assays; Autoimmune | Accident; Adolescent |
| Behavioral neurology; Bio; Biochemical; Biochemistry; Biological hematology; Biology; Burn | Biological |
| Cardiac; Cardiac Surgery; Cardiothoracic; Cardiothoracic anesthesiology; Cardiothoracic surgery; Cardiovascular; Cardiovascular surgery; Cellular; Cellular pathology; Central nervous system; Cerebrovascular; Cerebrovascular; Clinical biology; Clinical chemistry; Clinical immunology; Clinical laboratory sciences; Clinical microbiology; Clinical neurophysiology; Colon; Colon and Rectal Surgery; Colorectal; Colorectal surgery; Cosmetic; Cosmetic surgery; Craniofacial; Craniofacial surgery; Craniomaxillofacial; Craniomaxillofacial trauma | Cardiology; Care; Clinical; Critical care |
| Dental surgery; Dermatology; Dermatology–Venereology; Dietetics; Disaster; Disaster medicine | Diabetes; Diagnostic; Disabilities; Disease |
| Embryology; Emergency medicine; Endocrinology; ENT | Emergency; Emergency medical services |
| Facial cosmetics; Family; Family medicine; Fertility; Fertility medicine; Foot and ankle; Forefoot surgery; Forensic | |
| Gastro; Gastro-enterologic; Gastro-enterologic surgery; Gastroenterology; Gastrointestinal surgery; Genetics; Geriatric medicine; Geriatric neurology; Geriatrics; Gynecologic oncology; Gynecology | |
| Haematology; Hand surgery; Head and neck; Headache medicine; Hematology; Hepatology; Hospice and palliative medicine; Hospital medicine; Hyperbaric | Health; Health informatics; Hospital emergency |
| Child; Child and adolescent psychiatry and psychotherapy; Child psychiatry | Chemical |
| Immunology; Infectious; Infectious disease; Inflammatory diseases; Intensive care medicine; Interventional radiology | Intensive care; Internal medicine |
| Kidney; Kidney diseases | |
| Laboratory medicine; Laryngology | Laboratory |

**Table A1.** *Cont.*

| No | Yes |
| --- | --- |
| Maternal–fetal medicine; Maxillofacial surgery; Maxillo-facial surgery; Medical genetics; Medical toxicology; Microbiology; Microscope analysis; Microsurgery; Midfoot surgery; Mohs surgery; Musculoskeletal | Medical; Medical Model; Medical Simulation; Medicine; Medical research; Model |
| Neonatology; Nephrology; Nerve; Nervous system; Neuro; Neurodevelopmental disabilities; Neurological surgery; Neurology; Neuromuscular medicine; Neuro-oncology; Neuropsychiatry; Neuro-psychiatry; Neuroradiology; Neurosurgical oncology; Neurotology; Neurotrauma; Nuclear; Nuclear medicine; Nutrition | Neurosurgery |
| Obstetrics; Occupational; Occupational medicine; Ophthalmology; Oral; Oral surgery; Orthodontics; Orthopaedics; Orthopedic; Orthopedic surgery; Orthopedic trauma surgery; Otolaryngology; Otorhinolaryngology | Oncology |
| Paediatric; Paediatric allergology; Paediatric cardiology; Paediatric endocrinology and diabetes; Paediatric gastroenterology, hepatology and nutrition; Paediatric haematology and oncology; Paediatric infectious diseases; Paediatric nephrology; Paediatric respiratory medicine; Paediatric rheumatology; Paediatric surgery; Paediatrics; Pain medicine; Palliative care; Pediatric; Pediatric anesthesia; Pediatric cardiology; Pediatric emergency medicine; Pediatric endocrinology; Pediatric gastroenterology; Pediatric hematology; Pediatric neurosurgery; Pediatric oncology; Pediatric ophthalmology; Pediatric orthopedic surgery; Pediatric surgery; Peripheral; Peripheral nerve; Peripheral nervous system; Pharmacology; Physiatry; Physical medicine; Physical medicine and rehabilitation; Plastic; Plastic surgery; Podiatric Surgery; Podiatry; Preventive; Preventive medicine; Proctology; Psychiatry; Pulmonology | Pain; Pain management; Pathology; Public Health |
| Radiation; Radiation Oncology ; Radiology; Rear foot surgery; Reconstructive surgery; Rectal Surgery; Reproductive; Reproductive medicine; Respiratory; Respiratory medicine; Rheumatic diseases; Rheumatology | Rectal; Rehabilitation |
| Skull base; Sleep medicine; Spinal column; Spine surgery; Sports medicine; Stereotactic and functional; Stomatology; Surgical oncology; Surgical sports medicine | Simulation; Spine; Surgery; Surgical |
| Thoracic surgery; Toxicology; Transfusion; Transfusion medicine; Transplant; Transplant surgery; Trauma; Trauma care; Trauma surgery; Tropical medicine | Therapeutic |
| Ultrasound; Undersea; Undersea and hyperbaric medicine; Urgent; Urgent Care; Urgent Care Medicine; Urology | |
| Vascular medicine; Vascular neurology; Vascular; Vascular surgery; Venereology; Venerology | |
| Wilderness medicine | |

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
