# Peer review of "Specialization of Business Process Model and Notation Applications in Medicine—A Review"

_data, 2020_

Round 1

Reviewer 1 Report

This paper has now been improved from the previous submission with recent literature in the healthcare domain.

This review method is quite well described in the paper and the Kitchenham approach appears to me appropriate. As well, the Preferred Reporting Items for Systematic Reviews and Meta-analyses (PRISMA) and Methodological Expectations of Cochrane Intervention Reviews (MECIR) are relevant approaches.

Yet, as one major comment, the results are not sufficiently, commented, explained, analyzed for giving more value to the contribution of this paper.

The charts, maps and Mind Map of the BPMN application in medical specializations are a convenient representation to characterize a global view of the question.

Nevertheless, I still believe that it would be better to consider Google Scholar instead of WoS even if all the papers are not all evaluated. So the report about BPMN in HC is still not fully convincing me since the study is only on a subset of published papers.

In any case the paper has the merit to exist as one review of BPMN in HC. The paper gives a recent literature review about the use of BPMN in medical domain it can be interesting for research in this domain.

Some specific comments:

I’m surprised by the Figure 11 results; can you double check it?

The figure 16 – 20 have to cite and recall the period (years) they capture. As well, they are not very commented. What to conclude about the evolution regarding time?

Author Response

Comment 1: the results are not sufficiently, commented, explained, analyzed for giving more value to the contribution of this paper.

Answer 1: Thanks for the comment. We have expanded the texts describing the results of the analyzes.

Comment 2: it would be better to consider Google Scholar instead of WoS even if all the papers are not all evaluated. So the report about BPMN in HC is still not fully convincing me since the study is only on a subset of published papers.

Answer 2: Thanks for the comment. Yes, we agree that the results of this text are only a subset of all existing publications. Therefore, we emphasized this in the limits of the study. We have now added this section as the fourth section.

Comment 3: I’m surprised by the Figure 11 results; can you double check it?

Answer 3: Thanks for the comment. We checked the result; it's okay. It is the only one document published in 2007 which now has 105 citations. As it is the only document of this country, it is the best concerning average values of other countries.

Comment 4 : The figure 16 – 20 have to cite and recall the period (years) they capture.

Answer 4: Thanks for the comment. Labels now contain a period.

Reviewer 2 Report

The paper addresses a systematic review of the use of the BMPN notation in the domain of medicine. The objective raised by the authors is interesting and relevant. The PRISMA methodology for reporting in systematic reviews and meta-analyzes has been correctly followed. However, the manuscript is of poor quality. The main issues to be resolved are the following:

- It would be better to rethink the research questions as real questions and not as a starting hypothesis. As the study is outlined, perhaps the research questions should be something similar to: RQ1: in which medical specializations is the BMPN nation used?; RQ2; what types of use is made of the BMPN notation in these specialities?; RQ3: Have the different versions of the BPMN notation brought about a change in its use in the medical domain?

- The meta-analysis of the papers found in the literature should explicitly focus on answering the research questions raised. The need or convenience of analyzing, for example, the authors of the publications is not perceived when this could be compromising (since the information of the authors is very partial). It is interesting to analyze the temporal evolution of the use of the BPMN notation in the domain or even the geographical region in which the use of BPMN is circumscribed (although this can also be very partial).

- There is a lot of data in the manuscript, but little meaningful information. Data such as that presented as text between lines 259-284 or between lines 304-402 obfuscates the reading of the manuscript. Authors should find another way to represent these data (when it is of interest) and present its analysis in the text.

- Section 3.9 (research questions) is the heart of the study and should be expanded to present a more in-depth answer to the research questions posed.

- A section should be added in which the limitations of the study are presented.

The manuscript should also be revised by a native English speaker, as it presents many inaccuracies and, especially in the first pages, is very repetitive.

Author Response

Comment 1: It would be better to rethink the research questions as real questions and not as a starting hypothesis.

Answer 1: Thanks for the comment.  According to your recommendation, we reformulated the research questions into a real question.

Comment 2: The meta-analysis of the papers found in the literature should explicitly focus on answering the research questions raised.

Answer 2: Thanks for the comment. We have removed texts and results that were not relevant to the Research Questions.

Comment 3: There is a lot of data in the manuscript, but little meaningful information.

Answer 3: Thanks for the comment.  We rewrote the text of the publication to include more descriptive text and less raw data.

Comment 4: Section 3.9 (research questions) is the heart of the study and should be expanded to present a more in-depth answer to the research questions posed.

Answer 4: Thanks for the comment.  As we have expanded the previous text, which gradually answers the research questions, there is a section 3.9. just a summary and further extension of the text would lead to duplication.

Comment 5: A section should be added in which the limitations of the study are presented.

Answer 5: Thanks for the comment.  We have created a section 4 "The limitations of the study", where we describe and discuss the limitations of our study.

Comment 6: The manuscript should also be revised by a native English speaker

Answer 6: Thanks for the comment.  We have double-checked the text, but if you still consider the text to be of poor quality, we are ready to order paid proofreading services if the publication is accepted.

Round 2

Reviewer 1 Report

The comments have been considered. No more comment.

Author Response

Thank you so much for your time devoted to reviewing our article. Your valuable recommendations contribute to better text quality.

Reviewer 2 Report

The changes made by the authors have made the paper better focused. The manuscript presents an interesting work that has been developed and now presented in an appropriate way.

In any case, in my honest opinion, I think that the narrative of the manuscript could be improved, particularly in the first pages. A more elaborate and consistent description of the work done would significantly raise the quality of the paper.

Author Response

Thank you very much for your valuable comments throughout the review process. We have rearranged the text and edited it into a more readable form. Even so, we are ready to discuss the English proof-reading with the Editors.

This manuscript is a resubmission of an earlier submission. The following is a list of the peer review reports and author responses from that submission.

Round 1

Reviewer 1 Report

Summary:

In this paper, the author presents an overview about the application of Business Process Model and Notation (BPMN) in the medical field. Therefore, in the scope of two research questions ("In which areas of medicine were BPMN used?" and "What was the purpose of using BPMN?"), a systematic literature review was conducted. For the first question, 86 and, for the second one, a total of 38 articles were identified. The identified articles were analyzed and the conclusion of the author is that a BPMN supported-approach is only applied in about 15 % in the medical field.

General comments:

- The author provide relevant knowledge, background information, and motivations
- The research approach as well as respective information and concepts are described.
- Despite the presentation of above-mentioned points, a lot of relevant information about the approach of the author are missing (e.g., a replication is not possible). Further, it is not clear what processing-steps were taken in the review. More specifically, the used methodology is not adequately described.
- However, one of my main concern is that a lot of relevant research from literature is missing. For example, works addressing the mobile health sector are completely missing. The reasons are, on the one hand, that not enough sources were considered and, on the other, not appropriate search strings were used.
- The results of the literature reviews are presented in detail in the appendix.
- The authors take care of a common thread and respective sections are aligned accordingly.
- Different styles are used in the reference section. Moreover, the general presentation should be revised. There are, for example, some style issues.
- There are spelling errors and grammatical issues in the manuscript.
- Quality of figures should be improved (e.g., Fig. 3 is blurry)

Comments to Introduction section:

- The main intention/idea of the manuscript is deducible from the motivation. However, the presentation is not optimal. In my opinion, a clear presentation about the why and wherefore should be added to this introduction.
- There are statements in the paper that should be substantiated (e.g., Process management allows the processes to be modernized or systems reworked to support new clinical practices, regulatory standards, cost recovery methods, and the like).

Comments to Study section:

- In general, why was the first question defined as Pre-Q1? Since there is no Q1, only a follow-up question Q2, Pre-Q1 can be defined as Q1. The prefix suggest there is an actual Q1 and is therefore confusing.
- As already mentioned, one of my main concern is about the incomplete methodology and, as a result, this mandatory section should be revised critically. Information should be included such as:
- What were the search strings?
- Why were only two sources for publications considered? What about Google Scholar, IEEE, etc.?
- Why were both methods (i.e., PRISMA and MECIR) used? Was one insufficient?
- Regarding EC 1, why is this an exclusion criteria? With that definition, publications addressing, for example, the modeling of existing medical-related processes for the purpose of analysis and optimization would be excluded. In my opinion, due to the exclusion of such publications, the review of this work is not complete. As already mentioned, a lot of work in this context (e.g., mHealth) is missing.
- Further, it is unclear whether two (one for Pre-Q1 and Q2) or one review (Pre-Q1 and Q2 are interrelated) were/was conducted. Since this is a vital information having an impact on the understanding of the Result Section, it should be presented in a clear manner.

Comments to Result section:

- For me, it is not clear how the medical fields, in which BPMN was not applied, have been identified. Have you removed from the set of all medical fields the one that has already been covered by publications? If yes, I don't think this is a valid approach, since a lot of not considered information (e.g., mHealth) gets lost. In addition, in Section 2, Pre-Q1 addresses medical fields in which BPMN has been applied and now, in Section 3, Pre-Q1 addresses medical fields in which BPMN has not been applied. When did this negation happen? Further, how were the diagrams created?
- In general, please provide a summarizing table about the identified publications for a quick overview of the outcome. That would help the reading comprehension a lot.
- Since it is not clear, which aspect Pre-Q1 addresses, the statistical verification is not clear. In this context, what was the motivation of doing this statistics? Further, what test was applied? In general, since the two research questions are interrelated, it can be initially assumed that no significant differences can be found.

I really like the idea of this paper, since a comprehensive overview about the application of BPMN in the medical field is missing and, hence, would be an important contribution for future work. Therefore, I would like to encourage the author to stick with this issue, but to carefully revise the paper before another resubmission. The main reasons for my conclusion are that a lot of information about the methodology is missing and the analysis of the results is not clear. Further, and this is essential, the review is not complete and a lot of works (e.g., mHealth) were not considered in this review. Hence, concluding from my previous justifications.

Author Response

Summary:

In this paper, the author presents an overview about the application of Business Process Model and Notation (BPMN) in the medical field. Therefore, in the scope of two research questions ("In which areas of medicine were BPMN used?" and "What was the purpose of using BPMN?"), a systematic literature review was conducted. For the first question, 86 and, for the second one, a total of 38 articles were identified. The identified articles were analyzed and the conclusion of the author is that a BPMN supported-approach is only applied in about 15 % in the medical field.

General comments:

- The author provide relevant knowledge, background information, and motivations 

A1: Thank you for your assessment.

- The research approach as well as respective information and concepts are described.

A2: Thank you for your assessment.

- Despite the presentation of above-mentioned points, a lot of relevant information about the approach of the author are missing (e.g., a replication is not possible). Further, it is not clear what processing-steps were taken in the review. More specifically, the used methodology is not adequately described. 

A3: We have added a section on the methodology used, containing everything needed for possible replicability of the research. Besides, the research was updated to 10.6.2020. With this update, I have now conducted both studies in one.

- However, one of my main concern is that a lot of relevant research from literature is missing. For example, works addressing the mobile health sector are completely missing. The reasons are, on the one hand, that not enough sources were considered and, on the other, not appropriate search strings were used.

A4: The aim of this work is not an overall analysis of the health sector, but an analysis of medical specializations for the presence of the use of BPMN. Therefore, some crucial areas are missing, such as mHealth and others, which are written for general health care and care but are not specified for a specific medical specialization. The limit of this study is also the WOS database. This database was chosen as a guarantee of verified and quality information.

- The results of the literature reviews are presented in detail in the appendix.

A5: Thank you for your assessment. We have reworked the whole article and there is now only one table in the appendix.

- The authors take care of a common thread and respective sections are aligned accordingly.

A6: Thank you for your assessment.

- Different styles are used in the reference section. Moreover, the general presentation should be revised. There are, for example, some style issues. 

A7: This error was caused by internal database entries in BibTeX files. We went through and repaired all the items

- There are spelling errors and grammatical issues in the manuscript. 

A8: Thank you for your comment, we have corrected the entire text as part of the revision, if necessary we will have the final text corrected using the Proofreading service.

- Quality of figures should be improved (e.g., Fig. 3 is blurry)

A9: Thanks for the comment, we fixed and redesigned the figures.

Comments to Introduction section:

- The main intention/idea of the manuscript is deducible from the motivation. However, the presentation is not optimal. In my opinion, a clear presentation about the why and wherefore should be added to this introduction.

A10: Thank you for the comment. We rewrote the introduction and I hope it now provides a better presentation.

 - There are statements in the paper that should be substantiated (e.g., Process management allows the processes to be modernized or systems reworked to support new clinical practices, regulatory standards, cost recovery methods, and the like).

A11: Thank you, we have substantiated the statement and supported by the references.Comments to Study section:

- In general, why was the first question defined as Pre-Q1? Since there is no Q1, only a follow-up question Q2, Pre-Q1 can be defined as Q1. The prefix suggest there is an actual Q1 and is therefore confusing.

A12: Thanks for the comment, we updated the study and combined the results into a single study.

- As already mentioned, one of my main concern is about the incomplete methodology and, as a result, this mandatory section should be revised critically. Information should be included such as:
- What were the search strings?

A13: We have now introduced search queries in the text.

- Why were only two sources for publications considered? What about Google Scholar, IEEE, etc.?

A14: The WOS database was chosen as a guarantor of the quality of publications and as the most well-known field-wide database. I also searched for other databases, where only Scopus found more than 495 publications. Such a broad review was not in my power to be conducted during the major revision. Of course, I plan further research, which will contain more databases, but it will not focus on the analysis of medical specializations but in general on health care.

- Why were both methods (i.e., PRISMA and MECIR) used? Was one insufficient?

A15: The PRISMA and MECIR methods have been designed primarily for specific medical searches. Only the connection with Kitchenbaum was able to create a search on the border of the IT approach.

- Regarding EC 1, why is this an exclusion criteria? With that definition, publications addressing, for example, the modeling of existing medical-related processes for the purpose of analysis and optimization would be excluded. In my opinion, due to the exclusion of such publications, the review of this work is not complete. As already mentioned, a lot of work in this context (e.g., mHealth) is missing. 

A16: Thank you for the comment. We have considered all the exclusion criteria. There were only two publications excluded as review papers and they are listed in the Related Works section.

- Further, it is unclear whether two (one for Pre-Q1 and Q2) or one review (Pre-Q1 and Q2 are interrelated) were/was conducted. Since this is a vital information having an impact on the understanding of the Result Section, it should be presented in a clear manner. 

A17: Thank you for the comment. The whole study was performed (with current data and according to the recommendations of all reviewers) again and so there is only one study in the revised text, containing three research questions.

Comments to Result section:

- For me, it is not clear how the medical fields, in which BPMN was not applied, have been identified. Have you removed from the set of all medical fields the one that has already been covered by publications? If yes, I don't think this is a valid approach, since a lot of not considered information (e.g., mHealth) gets lost. In addition, in Section 2, Pre-Q1 addresses medical fields in which BPMN has been applied and now, in Section 3, Pre-Q1 addresses medical fields in which BPMN has not been applied. When did this negation happen? Further, how were the diagrams created? 

A18: Thanks for the comments. For each medical specialization, a search was performed in the database to see if there was a publication for the search string. As mentioned above, this was not a survey for the entire healthcare system, but a survey of specific medical specializations, which were selected from generally valid international documents.

Thank you for notifying me of the negation. In the current text, the concepts and procedures are already unified.

- In general, please provide a summarizing table about the identified publications for a quick overview of the outcome. That would help the reading comprehension a lot. 

A19: We have added a flow chart of the search process and all publications are listed with a specific search Queries in one table.

- Since it is not clear, which aspect Pre-Q1 addresses, the statistical verification is not clear. In this context, what was the motivation of doing this statistics? Further, what test was applied? In general, since the two research questions are interrelated, it can be initially assumed that no significant differences can be found.

A20: The part talking about statistics was removed on the recommendation of the reviewers, moreover, with the repetition and updating of the systematic review, it was no longer adequate.

I really like the idea of this paper, since a comprehensive overview about the application of BPMN in the medical field is missing and, hence, would be an important contribution for future work. Therefore, I would like to encourage the author to stick with this issue, but to carefully revise the paper before another resubmission. The main reasons for my conclusion are that a lot of information about the methodology is missing and the analysis of the results is not clear. Further, and this is essential, the review is not complete and a lot of works (e.g., mHealth) were not considered in this review. Hence, concluding from my previous justifications.

A21: Thank you very much for the comment, I hope that the changes made are sufficient for publication in this journal. However, the topic will be further explored on a much broader basis in a future publications.

Reviewer 2 Report

The paper addresses the study of the use of BPMN notation in the health field. Specifically, it tries to identify, on the one hand, medical specialities in which there are no proposals for the use of this notation and, on the other hand, to determine what purposes are given to this notation in the different sub-areas of health.

The manuscript is somewhat difficult to read (it should be reviewed by a native English speaker), but it is interesting, mainly in the part that synthesizes the uses that are given to the BPMN notation in the field of health. In my opinion, this synthesis is helpful to researchers in this domain.

Before publishing, the following issues should be improved:

- In some points of the manuscript, even in the abstract, a relationship is established between service-oriented architectures and BMPN. This relationship does not seem obvious and, therefore, should be explained.

- The content of section 1.1 should be merged directly with the contents of section 1.

- Why is the first research question identified with the Pre-Q1 identifier? Why not Q1 directly? This seems somewhat confusing. The first research question seems to have the same entity as Q2.

- The flow diagram of activities of the PRISMA guidelines should be included. At least for the second study.

- It would be useful to include the references of the articles in each branch of the mindmap.

- In my opinion, section 3.3 (statistical verification) should be removed.

Author Response

The paper addresses the study of the use of BPMN notation in the health field. Specifically, it tries to identify, on the one hand, medical specialities in which there are no proposals for the use of this notation and, on the other hand, to determine what purposes are given to this notation in the different sub-areas of health.

The manuscript is somewhat difficult to read (it should be reviewed by a native English speaker), but it is interesting, mainly in the part that synthesizes the uses that are given to the BPMN notation in the field of health. In my opinion, this synthesis is helpful to researchers in this domain.

A1 : Thank you for the comment.

Before publishing, the following issues should be improved:

- In some points of the manuscript, even in the abstract, a relationship is established between service-oriented architectures and BMPN. This relationship does not seem obvious and, therefore, should be explained.

A2: Thank you for your comment, the statement has been clarified in the text and supported by appropriate reference

- The content of section 1.1 should be merged directly with the contents of section 1.

- Why is the first research question identified with the Pre-Q1 identifier? Why not Q1 directly? This seems somewhat confusing. The first research question seems to have the same entity as Q2.

A3: It was a matter of distinguishing two systematic review, which took place at different times for different purposes on the same database with the same search strings. A new up-to-date systematic review was carried out in the revised text on the recommendation of reviewers. In this revision, the only one systematic review for the latest data is presented.

- The flow diagram of activities of the PRISMA guidelines should be included. At least for the second study.

A4: We added a flow chart.

- It would be useful to include the references of the articles in each branch of the mindmap.

A5: Thank you for your comment. We have created many analyzes for this text and replaced the Mind Map with a content analysis, including links to relevant publications.

- In my opinion, section 3.3 (statistical verification) should be removed.

A6: this section has been deleted.

Reviewer 3 Report

  There are many concerns that prevent me from recommending the publication of this article.

  • Regarding BPMN. I do not share the point of view of the authors regarding the linking between SOA and BPMN. It is true that SOA models are almost always based on BPMN models. But, BPMN does not require SOA, actually it does not require any technological support at all. BPMN just allows the definition of Business Process Models. And this is important in this context because many hospitals use BPMN just to describe how to implement medical protocols regarding, for instance, how to administrate parenteral drugs, for example. Also I would suggest to improve de organization of the introduction (it does not make sense to have just 1 single subsection)
  • Regarding the research question. I do not get the concept of “pre-study”. I would suggest that Pre-Q1 should just be Q1 or just be removed.
  • Regarding the PRISMA Methodology. I have many concerns about this point:
    • Why are not you including the PRISMA flow diagram? (http://www.prisma-statement.org/documents/PRISMA%202009%20flow%20diagram.doc). I think it is “almost” compulsory in this kind of analysis analysis.
    • Why not include the queries used? And the list of articles selected?
    • Also, I am unsure about Table A2. Which was the criteria for that selection? Why “Emergency medicine” (for example) is not included?
    • What is the difference between Pre-IC2 and IC2? Why not EC-2 applied to Pre-Q1? Any paper accepted on Pre-Q1 was excluded from Q2
  • About 3.3 Statistical verification. I am sorry but I fail to understand what you are comparing. Maybe it is just my fault, but could you explain why does the difference between Pre-q1 and q2 matter? Or how is that relevant?
  • About the conclusions:
    • Where/how did you get the areas for Pre-q1? I guess [36]… If so, the analysis should be “more straighfoward” and discussion about which are not used or why could be interesting. Even more, 85% may no be as relevant as some information about the relevance of those area and its impact.
    • On Figure 3 you mention in point 4 “Business processes in the healthcare sector”. That seems to be a bit vague. No information can be obtained.
  • Do the colors on figures 4 and 5 mean something? What’s the explanation to include PRISM on figure 5? How can these figures be interpreted/what information convey?

Minor details:

-Figures A1, A2 are cropped

-Page 7. Line 214. (CP) is missplaced

Author Response

There are many concerns that prevent me from recommending the publication of this article.

  • Regarding BPMN. I do not share the point of view of the authors regarding the linking between SOA and BPMN. It is true that SOA models are almost always based on BPMN models. But, BPMN does not require SOA, actually it does not require any technological support at all. BPMN just allows the definition of Business Process Models. And this is important in this context because many hospitals use BPMN just to describe how to implement medical protocols regarding, for instance, how to administrate parenteral drugs, for example. Also I would suggest to improve de organization of the introduction (it does not make sense to have just 1 single subsection)

A1: Thanks for the comment, the information about SOA and BPMN has been corrected. The Introduction section has been reworked

  • Regarding the research question. I do not get the concept of “pre-study”. I would suggest that Pre-Q1 should just be Q1 or just be removed.

A2: Thank you, after incorporating the recommendations of all reviewers, the study was repeated and now one study containing the best of the previous two is presented.

  • Regarding the PRISMA Methodology. I have many concerns about this point:

Why are not you including the PRISMA flow diagram? (http://www.prisma-statement.org/documents/PRISMA%202009%20flow%20diagram.doc). I think it is “almost” compulsory in this kind of analysis analysis.

A3: We added a flow chart.

  • Why not include the queries used? And the list of articles selected?

A4: We have provided Search Queries with corresponding publications..

  • Also, I am unsure about Table A2. Which was the criteria for that selection? Why “Emergency medicine” (for example) is not included?

A5: Thanks for the comment, you're right that without search strings, the table itself is confusing. I hope the new text is already in order.

  • What is the difference between Pre-IC2 and IC2? Why not EC-2 applied to Pre-Q1? Any paper accepted on Pre-Q1 was excluded from Q2

A6: Thank you for your comment, yes, the relationship between Pre-study and Study was not very well described, so after considering the comments of reviewers, we conducted the study repeatedly in one Systematic Search..

  • About 3.3 Statistical verification. I am sorry but I fail to understand what you are comparing. Maybe it is just my fault, but could you explain why does the difference between Pre-q1 and q2 matter? Or how is that relevant?

A7: This section has been removed from the revised version.

  • About the conclusions:
    • Where/how did you get the areas for Pre-q1? I guess [36]… If so, the analysis should be “more straighfoward” and discussion about which are not used or why could be interesting. Even more, 85% may no be as relevant as some information about the relevance of those area and its impact.

A8: Thanks for the comment. I hope the new text is already in order.

  •  
  • On Figure 3 you mention in point 4 “Business processes in the healthcare sector”. That seems to be a bit vague. No information can be obtained.
  • A9: Figure 3 has been reworked.
  •  
  • Do the colors on figures 4 and 5 mean something?
  • A10: Images have been reworked if the colors now have any meaning. is shown in the picture or caption.
  •  
  • What’s the explanation to include PRISM on figure 5?

A11: Thank you, I have added an explanation of the abbreviations used in the figures.

  • How can these figures be interpreted/what information convey?
  •  

A12: Thank you, a more detailed description and explanation of the figures has been added.

Minor details:

-Figures A1, A2 are cropped

A13: thank you figures have been reworked

-Page 7. Line 214. (CP) is missplaced

A14: thank you,it  has been fixed.

Reviewer 4 Report

This paper identifies and references medical specialties where the use of BPMN is not published yet (Pre-Q1) and the case it is used it lists for which medical specialties it is used (Q2).

This paper arrives in the healthcare domain, BPMN is already used frequently and combined with other approaches and methods. As one major combined approach, Simulation of process behavior appears in several existing contributions. I recommend to read and cite [1]

This paper gives a literature review about the use of BPMN in medical domain. This review method is quite well described in the paper and the Kitchenham approach is appropriate. As well, the Preferred Reporting Items for Systematic Reviews and Meta-analyses (PRISMA) and Methodological Expectations of Cochrane Intervention Reviews (MECIR) are relevant approaches. The Mind Map of the BPMN application in medical specializations is a convenient representation to characterize a global view of the question.

Nevertheless, the conclusion in the paper that for 85% of medical specialties, the use of BPMN has not been published yet is not fully convincing since the study is only on a subset of published papers. A lot of papers have been published somewhere else [2,3,4] that should have been considered. But, the choice for primary sources of data with publications listed only in the Web Of Science of Clarivate Analytics database is not capturing paper out of WoS and it is not motivated sufficiently why only WoS. It can be criticized first since the list is not publicly accessible. So, other major databases should be considered as well. It would be required to extend to Scopus as well as Google scholar. E.g. Healthcare Access in Rural Communities is represented in the literature and not discussed in the paper.

The end of the sentence is not clear for me: “the use of BPMN was not published yet (Pre-Q1) and the purpose of this connection (Q2).”

[1] De Ramón Fernández, A., Ruiz Fernández, D., & Sabuco García, Y. (2019). Business Process Management for optimizing clinical processes: A systematic literature review. Health Informatics Journal.

[2] Rojo, M. G., Rolón, E., Calahorra, L., García, F. Ó., Sánchez, R. P., Ruiz, F., ... & Espartero, R. M. (2008, December). Implementation of the Business Process Modelling Notation (BPMN) in the modelling of anatomic pathology processes. In Diagnostic pathology (Vol. 3, No. 1, p. S22). BioMed Central.

[3] Mariem Sbayou, Youssef Bouanan, Grégory Zacharewicz, Bruno Vallespir. BPMN Coordination And DEVS Network Architecture For Healthcare Organizations. International Journal of Privacy and Health Information Management, IGI Global, 2019, 7 (1), pp.103-115.

[4] Domingos, D., Respício, A., & Martinho, R. (2020). Reliability of IoT-aware BPMN healthcare processes. In Virtual and Mobile Healthcare: Breakthroughs in Research and Practice (pp. 793-821). IGI Global.

Author Response

This paper identifies and references medical specialties where the use of BPMN is not published yet (Pre-Q1) and the case it is used it lists for which medical specialties it is used (Q2).

This paper arrives in the healthcare domain, BPMN is already used frequently and combined with other approaches and methods. As one major combined approach, Simulation of process behavior appears in several existing contributions. I recommend to read and cite [1]

A1: Thank you for the comment. I have read the recommended publication and it is now incorporated into the text, including the citation.

This paper gives a literature review about the use of BPMN in medical domain. This review method is quite well described in the paper and the Kitchenham approach is appropriate. As well, the Preferred Reporting Items for Systematic Reviews and Meta-analyses (PRISMA) and Methodological Expectations of Cochrane Intervention Reviews (MECIR) are relevant approaches. The Mind Map of the BPMN application in medical specializations is a convenient representation to characterize a global view of the question.

A2Thank you so much for your review.

Nevertheless, the conclusion in the paper that for 85% of medical specialties, the use of BPMN has not been published yet is not fully convincing since the study is only on a subset of published papers. A lot of papers have been published somewhere else [2,3,4] that should have been considered. But, the choice for primary sources of data with publications listed only in the Web Of Science of Clarivate Analytics database is not capturing paper out of WoS and it is not motivated sufficiently why only WoS. It can be criticized first since the list is not publicly accessible. So, other major databases should be considered as well. It would be required to extend to Scopus as well as Google scholar. E.g. Healthcare Access in Rural Communities is represented in the literature and not discussed in the paper.

A3: thanks for the comments. The WOS database was chosen as a guarantor of the quality of publications. For example, the Google Scholar database contains many materials that have not been complied by any review process. Of course, the WOS database is not the only suitable source of evidence for systematic search. For this publication, however, I decided to focus only on this database and because of awareness of this database in scientific circles. I considered expanding the systematic search with additional databases and decided that it was not in my power to conduct such an extensive search as part of a major revision of the article. In my future work, however, I intend to address this topic across various databases for more general concepts than the currently selected specializations. Unfortunately, I deal with the topic individually, so if the reviewer knows about a possible collaborator, I will be happy to establish cooperation.

The end of the sentence is not clear for me: “the use of BPMN was not published yet (Pre-Q1) and the purpose of this connection (Q2).”

A4: Thank you, the sentence has been corrected as part of the overall stylistic revision and text changes.